# Exosomes: Their Role in Pathogenesis, Diagnosis and Treatment of Diseases

**DOI:** 10.3390/cancers13010084

**Published:** 2020-12-30

**Authors:** Houssam Aheget, Loubna Mazini, Francisco Martin, Boutaïna Belqat, Juan Antonio Marchal, Karim Benabdellah

**Affiliations:** 1GENYO Centre for Genomics and Oncological Research, Genomic Medicine Department, Pfizer/University of Granada/Andalusian Regional Government, Health Sciences Technology Park, Av. de la Illustration 114, 18016 Granada, Spain; francisco.martin@genyo.es; 2Department of Biology, Faculty of Sciences, University Abdelmalek Essaâdi, Tétouan 93000, Morocco; bbelqat@uae.ac.ma; 3Center of Biological and Medical Sciences (CIAM), Mohammed VI Polytechnic University, Ben-Guerir 43152, Morocco; loubna.MAZINI@um6p.ma; 4Biomedical Research Institute (ibs. GRANADA), 18012 Granada, Spain; jmarchal@ugr.es; 5Biopathology and Regenerative Medicine Institute (IBIMER), Centre for Biomedical Research (CIBM), University of Granada, 18016 Granada, Spain; 6Department of Human Anatomy and Embryology, Faculty of Medicine, University of Granada, 18016 Granada, Spain; 7Excellence Research Unit Modeling Nature (MNat), University of Granada, 18016 Granada, Spain

**Keywords:** exosomes, molecular composition, cancer pathogenesis, diagnostics, therapeutics

## Abstract

**Simple Summary:**

The aim of this review is to provide an overview of the current scientific evidence concerning the role played by exosomes in the pathogenesis, diagnosis and treatment of diseases. The potential use of exosomes as delivery vectors for small-molecule therapeutic agents will be discussed. In addition, a special emphasis will be placed on the involvement of exosomes in oncological diseases, as well as to their potential therapeutic application as liquid biopsy tools mainly in cancer diagnosis. A better understanding of exosome biology could improve the results of clinical interventions using exosomes as therapeutic agents.

**Abstract:**

Exosomes are lipid bilayer particles released from cells into their surrounding environment. These vesicles are mediators of near and long-distance intercellular communication and affect various aspects of cell biology. In addition to their biological function, they play an increasingly important role both in diagnosis and as therapeutic agents. In this paper, we review recent literature related to the molecular composition of exosomes, paying special attention to their role in pathogenesis, along with their application as biomarkers and as therapeutic tools. In this context, we analyze the potential use of exosomes in biomedicine, as well as the limitations that preclude their wider application.

## 1. Introduction

Membrane-bound and heterogeneous extracellular vesicles (EVs) were initially considered anecdotal examples of cell debris or apoptotic bodies released by the majority of cells [1]. EVs are now regarded as key diagnostic tools [2,3,4] and therapeutic agents [5]. EVs facilitate communication processes between near and distant cells. In addition, these vesicles can be grouped into two major categories: (a) microvesicles (MVs; 100–1000 nm), considered to be functional liposomes composed of molecules such as nucleic acids, proteins and functional lipids surrounded by a lipid bilayer and (b) exosomes (EXOs; 30–150 nm) (Figure 1) [6], which differ from MVs in their size, protein composition, buoyant density, release mechanism and potential physiological role [7,8,9,10]. In this review, we will focus mainly on exosomes, with particular emphasis on their composition. We will discuss their potential role in signaling under both physiological and different pathological conditions. Special attention will be paid to the therapeutic role of exosomes as delivery vectors, as well as their potential use as biomarkers and in clinical interventions.

## 2. Pathological Functions of Exosomes

Exosomes are known to transfer bioactive cargo between donor and recipient cells, ensuring pleiotropic functions in intercellular communication. They are also considered to be an important factor in tumor pathogenesis and immunosuppression [14]. They generate an intricate network of interactions that inhibit the immune system by delivering similar contents of tumor cells to immune cells and also impair natural killer cell activation and induce effector T cell apoptosis [15]. These vesicles have been reported to use autocrine and paracrine signaling pathways to regulate cell characteristics, to modulate their microenvironment and to boost their effects [16]. In addition, exosomes can act as external stimuli and modify the biological phenotype of recipient cells by changing their RNA expression and activating their receptors. Interestingly, cancer cells exchange exosomes with stromal cells in order to create a protumor microenvironment and to increase tumor invasion and proliferation [17].

On the other hand, these vesicles facilitate the interneuronal transmission of pathogenic proteins that are responsible for several neurodegenerative diseases, such as Parkinson’s disease (PD) and Alzheimer’s disease (AD) [18]. The exosomal transfer of p-tau and Aβ1-42 between cells and body fluids is potentially involved in the slow progress of AD. Moreover, early detection of these neurodegenerative proteins could lead to successful treatments and longer survival [19]. Thus, the key protein involved in PD pathology α-synuclein is secreted via a calcium-dependent mechanism and transported by exosomes, leading to cell death in recipient cells [20]. Exosomes have also been reported to release cellular prion protein (PrPc) and prion protein scrapie (PrPsc) to the extracellular environment, thereby contributing to the pathological spread of infectious prions [21].

### 2.1. Tumor Pathogenesis

Tumor cells influence both their surrounding microenvironment and distant organs where they can promote angiogenesis, proliferation and cancer metastasis. Exosomes, which are considerably involved in cancer growth and metastatic spread, are considered the main cause of the paracrine effect on recipient cells (Figure 2). The regulation of oncogene expression and abnormal transformations might also result from different initiation factor effects. Eukaryotic translation initiation factor 3 (elF3) bridges the 43S preinitiation complex and elF4F-bound mRNA to control protein synthesis, and their aberrant expressed subunits are associated with different cancers [22]. The transforming growth factor beta (TGF-β) signaling pathway, another cancer initiation and progression factor, acts through its central mediator SMAD4 by disrupting DNA damage responses and repair mechanisms, thus enhancing their genomic instability [23]. This signaling is also targeted by the migration inhibitory factor (MIF) to induce the fibronectin production necessary for the remodeling of the premetastatic niche [24]. Additionally, TGF-β is reported to increase fibroblast growth factor-2 (FGF-2) production and mesenchymal stem cell (MSC) differentiation into myofibroblasts to trigger cancer proliferation and invasiveness [25,26]. In the tumor environment, the production of hypoxia inducible factor-1 (HIF-1) plays a crucial role in cancer initiation and progression. Consequently, hypoxia induces HIF-1 stabilization, and its nuclear translocation fosters the expression of genes such as vascular endothelial growth factor (VEGF), hepatocyte growth factor (HGF) and the Met protooncogene [27]. The oncogenes Kristen rat sarcoma 2 viral oncogene homolog (KRAS), epidermal growth factor (EGF) and SRC are transferred by exosomes to recipient tumor cells to promote tumor invasion [28]. To ensure the tumor evasion of immune surveillance, exosomes also release programmed death-ligand 1 (PD-L1) [29].

Heat shock proteins (HSPs), which are associated with stress conditions, are key regulators of exosome formation and release [30,31] and are involved in antitumor activity in a murine model in a major histocompatibility complex (MHC)-independent manner [32]. Furthermore, the P53 protein is mutated or lost in the majority of cancer types and also modulates many surveillance pathways [33,34]. This protein modulates the transcription of different genes, including TSAP6 and CHMP4C, thus promoting exosome production [35]. These transcriptional signals are involved in cell communication and immune activation [36]. Another tumor suppresser, the phosphatase and tensin homolog (PTEN) protein secreted in exosomes, presents phosphatase activity in target cells, resulting in the activation of the apoptosis cascade and suppression of cell proliferation through interactions with Notch signaling [37,38].

Exosomes containing noncoding RNAs (long noncoding and microRNAs (lnc- and miRNAs)) are associated with many cellular mechanisms [39,40]. MiRNAs were first identified in human serum and later in biological fluids such as saliva, urine and breast milk, thus confirming their role in cell-to-cell communication [7,41]. By modulating mRNA translation in target cells, exosome-associated miRNAs can improve and suppress cellular unbalance, development and tumorigenesis [42].

MiRNA secreted by nontumor cells can affect various cancer-associated mechanisms. Tumors not only contain cancerous cells but also vascular, immune and cancer-associated fibroblastic cells, as well as an extracellular matrix (ECM), all nontumor cells involved in cellular communication and signaling and sustain neighboring tumor cell growth and metastasis [43]. These normal cells secrete tumor-suppressive miRNAs in their EVs to competitively overcome the anarchic growth of their neighbors, a system failure that might initiate cancer [44]. This was observed in prostate cancer, where miRNA-143 acts as a death signal, inducing growth inhibition through a cell-competitive process [45]. Table 1 summarizes the commonly reported miRNAs and lnc-miRNAs found in cancer pathogenesis. Cancer is a multifactorial process in which different miRNAs are secreted by different cells belonging to the tumor microenvironment, resulting in intercellular communication and a single pathway, causing initiation and progression of the disease, angiogenesis, metastasis and drug resistance. In contrast, a single miRNA could be a key modulator of different signaling systems in multiple intercellular networks in recipient cells, thereby modifying their destination and signaling pathway, thus promoting tumorigenesis. Lnc-RNAs, which are highly expressed in exosomes, play a crucial role in the microenvironment by transferring cell signaling and by modulating gene expression [46,47].

#### 2.1.1. Cancer Initiation

Cancer is a genetic and irreversible change due to the activation of specific oncogenes, inactivation of tumor-suppressive genes or other genes involved in genome stability. The evolution of these cancer cells is the result of dual interactions between cancer cells and their surrounding microenvironments. Inflammation is considered the driving initiator of tumor development. Exosome integrins are reported to upregulate S100 proinflammatory molecules, probably by activating and phosphorylating SRCs [48]. Additionally, tumor cells induce the secretion of inflammatory factors, including VEGF, tumor necrosis factor-α (TNF-α), TGF-β and interleukins, to stimulate myeloid cells and immune cells to migrate, thus amplifying inflammatory factor secretion [49]. The immune response is prevented later after the programmed death receptor (PD-1) expressed on activated T and B cells and macrophages binds to its ligand, PD-L1, inducing T-cell apoptosis and inhibiting T-cell activation and proliferation [50,51]. Tumor-associated macrophages (TAMs), T-regulators and myeloid-derived suppressor cells are also recruited to the tumor to inhibit the immune response [52,53]. This immune suppression phase is followed by the improvement of angiogenesis and vascular permeability. In this case, MSCs and endothelial cell interactions mediated by Akt phosphorylation lead to the formation of a vascular microenvironment [54]. By expressing E-cadherin and carbonic anhydrase-9 (CA-9) on their surfaces, exosomes also promote angiogenesis [55,56]. Additionally, integrins (ITGs) present on their surfaces determine organotropism, and their different expressions are organ-specific [48]. These ITGs colocalize specifically with ECM components (laminin or fibronectin) whose composition is modulated by fibroblasts and endothelial cells, suggesting that exosomes drive the colonization of tumor cells by remodeling the stromal microenvironment of the target organ. Mesenchymal stem cells (MSCs) are part of the tumor microenvironment [57], where they are educated and transformed through the release of exosomes into tumor-supportive myofibroblasts, leading to cancer initiation [58].

Likewise, cancer cell-derived exosomes from multiple myeloma (MM) cells are reported to transfer miRNAs to MSCs to initiate cancer, which, in turn, activates cytokine secretion, tumor growth and migration [59]. This mutual intercellular communication is of primordial importance in initiating tumorigenesis in different organs. Tumor cells can also inhibit or decrease antitumorigenic miRNA activity, leading to cancer initiation [60]. The release of miRNA-202-3p by exosomes into the microenvironment negatively regulates its antitumorigenic target [61] (Table 1). From an alternative perspective, cancer-associated fibroblasts (CAFs), which are mostly present in the cancer microenvironment, could induce tumor development and progression. These cells, which secrete miRNAs such as miRNA-21, miRNA-378e and miRNA-143, significantly increase the stemness of breast cancer cells and their epithelial–mesenchymal phenotype [62]. In addition, infiltrating monocytes play an important role in tumor cell progression, as they are driven to differentiate into M2 tumor-associated macrophages (TAMs) by the derived exosomes miRNA-103a and miRNA-203, leading to the secretion of fibroblastic and proangiogenic growth factors [63,64].

On the other hand, exosome lncRNA-p21 is reported to suppress prostate cancer initiation and the expression of genes transcriptionally regulated by P53 [65]. P53 expression is also stimulated by lnc-RNA-MEG3 to inhibit cell proliferation in lung cancer [66]. Lnc-RNA-GAS5 represses antiapoptotic genes when binding to the DNA-binding domain of the glucocorticoid receptor to prevent prostate cancer initiation [67].

Other lnc-RNAs are reported to favor tumor progression by regulating or silencing different miRNAs involved in cancer initiation repression. LncRNA-HOTAIR, which is associated with poor prognosis in urothelial bladder cancer, sponges miRNA-205, thus facilitating tumor initiation and progression [68]. Similarly, lncRNA-MALAT1 is reported to modulate EMT and to promote cervical cancer cell growth and invasion [69,70]. LncRNA-MONC and MIR100HG are both expressed in acute megakaryoblastic leukemia blasts and act as oncogenes associated with tumor development [71]. LncRNA-RoR is a stress-responsive lncRNA in hepatocellular cancer, preventing the activation of cellular stress and miRNA-145 sponging, which can also promote the expression of hypoxia-inducible genes associated with cell growth, apoptosis, angiogenesis, differentiation and survival [72]. Another lnc-RNA, lncRNA DANCR, has been reported to sponge miRNA-33a-5p and to increase EMT, cell proliferation and migration in gliomas [73].

#### 2.1.2. Tumor Angiogenesis

The formation of tumor-associated vessels might be mediated by the sprouting of tumors surrounding pre-existing vessels or the newly recruited endothelial progenitor cells from bone marrow [60]. Exosome-derived miRNA-21 is reported to increase vascular endothelial growth factor (VEGF) levels (Table 1), the key player in angiogenesis, which facilitates endothelial cell migration and new blood vessel formation [74,75]. STAT proteins are also targeted by miRNA-9, whereby tumor neovascularization is strongly activated [76,77,78]. Another miRNA, miRNA-135b, transferred to endothelial cells by multiple myeloma cell-derived exosomes, inhibits hypoxia-inducible factor 1 (HIF-1) and promotes angiogenesis [79].

Angiogenesis is an important mediator of tumor progression through the induction of protumoral TAMs when monocytes incorporate miRNA-203-derived exosomes secreted by colorectal cancer cells [63] and miRNA-103a-derived exosomes from lung cancer [64]. This mechanism underlies the spread of cancer through the polarization of tumor-suppressive and proangiogenic macrophages. Exosomes also mediate the endothelial cell phenotype in CD90+ liver cancer cells through lnc-RNA H19 and promote angiogenesis and cell-to-cell adhesion [80].

#### 2.1.3. Tumor Metastasis

Since 1989, when Steven Paget introduced the concept of “seed and soil” in relation to tumor progression and metastasis, a great body of literature has been developed, with a better understanding of the mechanisms underlying tumor growth and metastasis [81]. The spread of tumor cells was proposed as the result of the interaction and cooperation between cancer cells (seed) and the host organ (soil) [82]. The metastatic process was later identified as including several stages, such as intravasation, extravasation, tumor latency and the development of micrometastasis and macrometastasis. However, the preferential target organs (soil) may be prepared for metastatic deposits through the development of premetastatic niches that facilitate tumor cell homing, colonization and growth. The primary tumor (seed) plays a key role in the development of premetastatic niches by producing soluble factors, inducing bone marrow-derived hematopoietic cell migration to the premetastatic niche. The primary tumor also secretes exosomes, thus modulating the tumor microenvironment in the premetastatic niches. EMT and mesenchymal-to-epithelial transition (MET) enable migratory phenotypes and seed behaviors. EMT enables tumor cells to enter the circulation and seeding at distant sites [83], where MET is responsible for colonization and metastasis [84]. Moreover, premetastatic niche formation is associated with the composition of molecular and cellular components undergoing four stages to support tumor growth and metastasis. In the primary phase, the primary tumor, which is affected by the uncontrolled proliferation and secretion of exosomes or other tumor-derived secreted factors (TDSFs), becomes hypoxic and inflammatory. Bone marrow-derived (BMD) immune/suppressive cells are prepared and mobilized to form an immature premetastatic niche at a distant organ or at another site of the same organ [85]. In the second licensing phase, BMDCs are continuously recruited in the secondary site in response to exosomes and TDSFs, and their interactions with the distant microenvironment lead to their maturation and preparation for tumor cell colonization. Apart from these BMDCs, bone marrow mesenchymal stem cells (BM-MSCs), which have been identified in different studies, are recruited by the evolving tumor microenvironment as a major source of cancer-associated fibroblasts (CAFs) that boost tumor cell survival [86,87,88]. The activation of integrins, chemokines and the ECM plays a key role in this organotropism by enabling seeding and colonization in the secondary licensed site [48]. ECM remodeling, as well as the presence of interleukin-1 β (IL-1β) and myeloid-derived suppresser cells, result in the EMT profile of tumor cells [89,90]. The mature and fertile premetastatic niche is colonized by the tumor cells that can undergo latency if the niche microenvironment is not yet suitable during the initiation phase. In the case of a well-prepared niche, seeding and colonization with tumor cells lead to the formation of micrometastases. In the final progression phase, premetastatic niche hosting and support of more migratory tumor cells induce growth, expansion and progression to form macrometastases.

From another perspective, cancer stem cells (CSCs), also known as cancer-initiating cells, have the ability to self-renew and to regenerate the different cell subpopulations constituting the tumor [91], with evidence showing that few tumor cells can form a tumor and accomplish metastasis [91,92]. CSCs from metastatic breast cancer show significantly higher tumorigenic and metastatic capacities than low metastatic cells [93]. Althogh autophagy, whose contribution to tumor progression and metastasis remains controversial is considered to be another seed-type factor, some evidence has demonstrated its involvement in tumor invasion, colonization [94,95], in EMT [96] and CSC viability [95,97,98]. Tumor cells can also disseminate and metastasize in distant sites; however, a lag between both these processes can occur, with tumor cells entering a dormant state for long periods before giving rise to metastasis months or several years after the primary tumor treatment [99]. When these residual tumor cells, whose reactivation appears to be regulated by microenvironmental factors in certain organs, enter a dormant state, they become immune to therapy.

According to Paget, soil factors may first be represented by the primary tumor microenvironment and some molecules providing primary seed-to-soil signaling to enhance the invasive properties of tumor cells [100,101]. In different cancers, TAMs have been shown to induce tumor cell invasiveness through exosome-derived oncogenic miRNA-223, CCL18 and CCL19 [102,103]. MSCs promote cell motility through CCL5 signaling and endothelial cells by modulating oxygenation and tumor perfusion [104]. Besides promoting tumor growth and angiogenesis, CAFs also secrete SDF-1 to induce tumor cell motility and invasion [105]. Additionally, secondary soil, which plays a critical role in influencing cancer metastasis, is composed of many factors and cell types in the metastatic environment (distant organ microenvironment). In each cancer type, microenvironment-derived factors promote specific signaling, leading to tumor migration, cell adhesion, growth and metastasis by enabling tumor cells to enter the niche.

Invasive features are commonly associated with morphological changes in EMT migration, cytoskeleton organization, motility, the basal membrane and extracellular matrix (ECM) remodeling. Exosomes have emerged as potential regulators of the EMT promotion of tumor invasion and spread. Given that EMT is reversible, mesenchymal-to-epithelial transition (MET) might enable cancer cells to adopt an epithelial profile and capacity and, thus, transmigrate to distant sites, promoting metastasis [106]. The miRNA-200 family (miRNA-200a, -200b, -200c, -429 and -141) has the ability to regulate this epithelial cancer cell phenotype by inhibiting the expression of Zeb1 and Zeb2 gene repressors [107,108]. Being the principal component of the tumor microenvironment, fibroblasts play a crucial role in tumor progression. Their reprogramming into cancer-associated fibroblasts (CAFs) occurs after miRNA-105 and miRNA-155 induction in breast cancer and pancreatic cancer, respectively [109,110].

In addition, exosomes carrying different miRNAs have been shown to display migratory and metastatic behaviors leading to distant tumors [111]. By disrupting the vascular endothelial barrier, miRNA-939 and miRNA-105 are reported to increase its permeability through the VE-cadherin gene and by targeting the tight junction protein ZO-1, respectively [112,113]. In exosomes derived from breast cancer, miRNA-10b, with its higher enrichment levels, also promotes cell invasion [114]. The blood–brain barrier (BBB) is another aspect of tumor cell invasion, in which the modulation of permeability is the key feature of brain metastasis. BBB degradation is caused by miRNA-181c, which downregulates PDPK1 gene expression [115].

Glucose uptake suppression by nontumor cells has also been reported to increase nutrient availability in the premetastatic niche via high-secretion miRNA-122, as observed in breast cancer patients with metastatic progression [116,117].

#### 2.1.4. Tumor Immunity

Exosomes have been reported to regulate adaptive immunity in different organs through the cytokines and miRNAs they secrete [118]. Their involvement in tumor immunity can range from the regulation of tumor antigens to tumor immunity polarization [119,120]. However, the most commonly reported involvement of exosomes in immune responses relates to antitumor supportive activity and to their role in preventing immune surveillance. Tumor exosomes inhibit bone marrow dendritic cell (DC) differentiation via the modulation of interleukin-6 (IL-6) expression [121]. The regulatory factor-X-associated protein (RFXAP), a key transcription factor for the MHC-II gene, is downregulated by pancreatic cancer-secreted exosomes containing miRNA-212-3p, leading to the inhibition of MHC class II expression and CD4+ T-cell inactivation [122]. On the other hand, T-cell apoptosis can be induced via the Fas ligand [123], while cytotoxic natural killer (NK) cell activity can be inhibited via the downregulation of NK group 2 member D by tumor exosomes [124]. Regulatory T cells are induced by exosome-derived transforming growth factor β-1 (TGF-β) or miRNA-214 in order to downregulate the phosphatase and tensin homolog (PTEN) and to increase IL-10 secretion, leading to tumor growth [125,126].

On the other hand, tumor cells can evade immunosuppression responses by upregulating the surface expression of PD-L1 and by inactivating T cells. After binding to its receptor PD-1, the Sh2p-driven dephosphorylation of the T cell receptor and its coreceptor CD28 occurs, resulting in the suppression of the antigen-driven activation of T cells [127]. The level of PD-L1 in blood cancer patients is related to their pathoclinical features. Poggio et al. have also demonstrated the differential expression of exosomal PD-L1 in prostate cancer and melanoma cell lines [128].

Cancer cells release exosomes expressing PD-L1, which binds PD-1 through its extracellular domain on CD8 T cells in a concentration-dependent manner [53,128,129,130]. This PD-L1 secretion can be significantly amplified in tumor cells and in exosomes in response to interferon gamma (IFN-γ) [128,131]. Exosomal PD-L1 levels, which correlate with tumor size, have been reported to be significantly higher in the plasma of melanoma patients as compared to healthy donors. Breast and lung cancer cells also exhibit immunosuppressive exosomal PD-L1. Physical interactions were identified with exosomal PD-L1 and activated PD-1+ CD8 T cells, leading to the inhibition of their proliferation by reducing the expression of Ki-67 and Granzyme B, cytokine production and cytotoxicity through the inhibition of IFN-γ, IL-2 and TNF-α [129]. Using a preclinical model of prostate cancer, the TRAMP-C2 model, the cluster regulatory interspaced short palindromic repeats (CRISPR)/Cas9-mediated deletion of *Rab27a* and *PD1l*, thus inducing exosomal PD-L1 loss, has proven that exosomal PD-L1 is involved in in vivo tumor growth, even at distant sites [128]. Additionally, in the absence of exosomes or PD-L1, the CD8 T-cell fraction increases in lymph nodes relative to wild-type animals and decreases the exhaustion marker Tim 3 characterizing cell subpopulations and increases the Granzyme B marker. Thus, exposure to exosomal PD-L1-deficient tumor cells or the use of anti-PD-L1 antibodies, considered to be new antitumor therapeutic targets, suppresses tumor growth. Moreover, antibodies against PD-L1 and PD-1 have been demonstrated to be efficient in treating many cancer types.

Known to express different toll-like receptors (TLRs), DCs and MSCs are expected to interact with miRNAs to modulate immunity under normal and tumor conditions. Tumor exosomes release miRNA-21 and miRNA-29a, which are considered TLR family ligands in immune cells and act as key regulators of immune responses associated with prometastatic microenvironments [132]. Pancreatic cancer-derived exosomes regulate TLR4 secretion and the production of cytokines such as TNF-α and IL-12 in DCs through miRNA-203 transfer [133]. DC-derived exosomes are reported to activate T and B cells, thus facilitating the presentation of tumor antigens released by cancer cell-derived exosomes [134]. Additionally, this activation of T and B cells might be amplified by mast cells when DC differentiation is induced [135].

#### 2.1.5. Cancer Drug Resistance

Tumor cells often display resistance, hampering tumor treatments aimed at decrease inter- and intracellular drug concentrations. This resistance can be the result of different mechanisms due to genetic or phenotypic changes termed intrinsic resistance or to extrinsic resistance involving the effect of the tumor microenvironment (TME) [136]. In the TME, endothelial cells, fibroblasts and immune cells interact to support tumor growth and progression, where homotypic or heterotypic exosome transfers are regarded as potent effectors [136,137,138].

Tumor cells presenting cancer predisposition display multidrug resistance (MDR), which is related to the increase in the expression of drug transporters from the adenosine triphosphate (ATP)-binding cassette transporter (ABC) family [139]. These transporters are present in more than 50% of cancer-presenting MDR phenotypes or can be induced by chemotherapy [140] and encoded by multidrug resistance protein 1 gene (*MDR1* or *ABCB1*) for the p-glycoprotein or the *ABCG2* gene for the breast cancer resistant protein (BCRP) [141]. Additionally, these transporters are able to transfer drug resistance through exosomes to sensitive cells [142,143,144]. On the other hand, by reversing their orientation in the exosome membrane, the transporters can drive drugs from donor cells into exosomes for sequestration [143,144,145]. Acidification of the tumor microenvironment appears to promote drug sequestration by increasing the expression of H+-ATPases [146]. Exosomes can also act as sponges by presenting on their surface bait targets for drug molecules such as CD20 to trap the anti-CD20 rituximab [147].

Exosomes are also reported to mediate irradiation resistance by interacting with the cell cycle and DNA repair. Stroma-derived exosomes are reported to induce tumor cell dormancy through their recruitment in the G0 phase and a CSC phenotype, thus increasing chemoresistance [148]. When exosomes were derived from MSCs, a CSC phenotype was improved in tumor cells [149,150]. Exosomes can also mediate antiapoptosis in donor cells by decreasing the intracellular levels of proapoptotic proteins by releasing caspase-3 and -9 [151,152]. Besides decreasing these proapoptotic proteins, exosomes prevent apoptosis in recipient cells by stimulating antiapoptotic pathways mediated by IL-6, CD41, p38 and p53 and JNK, Raf/MEK/ERK and Akt [152,153,154]. IL-6, activin A and granulocyte-colony stimulating factor (G-CSF) have been shown to induce a CSC phenotype in lung carcinoma cells by stimulating their de-differentiation [155].

Inducing DNA damage repair is triggered by exosomes to induce tumor cell survival after exposure to genotoxic irradiation. Furthermore, irradiation increases tumor cell exosome release [156]. In breast cancer exosomes, the phosphorylation of ataxia telangiectasia mutated (ATM) kinase, Histone H2AX and checkpoint kinase 1 (ChK1) increases in recipient cells, leading to DNA damage repair responses [157]. DNA double-stranded break repair, induced by tumor cell exosomes to increase irradiation therapy, can occur in response to irradiation [156,157,158]. Exosomes derived from irradiated tumor cells can adopt a migratory profile to escape from the irradiated site, leading to an increase in irradiation resistance [159].

Cancer-associated fibroblasts (CAFs), which are largely regarded as the principal component of tumors and supportive cells, provide a nursing niche and actively regulate the survival and proliferation of cancer cells [137,138]. CAFs affect cross-interactions between the stroma and tumor to activate tumor-supportive mechanisms [160,161]. One of these mechanisms is related to the decrease in drug penetrance in the tumor microenvironment due to a desmoplastic reaction [162]. After exposure to chemotherapy, CAFs contribute to therapy resistance through the significant increase in exosome release. In response to gemcitabine exposure, these exosomes increase the chemoresistance-inducing factor SNAIL in recipient epithelial cells, leading to proliferation and resistance of pancreatic ductal adenocarcinoma [163]. In breast cancer, fibroblast-derived exosomes induce a CSC phenotype through Notch3/STAT1 [164], where, in lung cancer, these fibroblasts create a nursing microenvironment around aldehyde dehydrogenase 1-positive CSCs to resist chemotherapy [165].

Therapy resistance mediated by the CSC phenotype is closely related to EMT. Exosomes are actually regarded as the main inducers of EMT [166,167], and cross-interactions between EMT, CSCs, resistance and exosomes appear to be responsible for increasing CSC markers in breast cancer biopsies after chemotherapy [168]. Moreover, this EMT confers cell plasticity on CSCs and CAFs. However, CAFs and CAF-like phenotypes may release cancer-supportive signals after exposure to different chemotherapies, as well as to a single ablative dose of radiotherapy [138,161,169]. 

Increasing evidence demonstrates that miRNA-derived exosomes are involved in drug resistance in different cancers. Breast cancer exosome-derived miRNA-221/222 has been reported to increase tamoxifen resistance by reducing the target gene expression of P27 and Era [170]. Transferred by monocytes, miRNA-155 has been reported to target telomerase activity and telomere length through TERF1 in neuroblastoma cells, leading to enhanced chemotherapy resistance. The authors cited above also report that miRNA-21 is involved in ovarian cancer chemoresistance, which suppresses cell apoptosis by binding to its target, APAF1 [171]. In addition, multidrug resistance protein 1 (MRP-1) is reported to be overexpressed in the promyelocytic leukemia HL60 cell line [172]. Nevertheless, cancer cells might target other adaptation mechanisms to escape chemotherapy; for example, in breast cancer, exosome-derived miRNA-9-5p, miRNA-195-5p and miRNA-203a-3p trigger the expression of stemness-associated genes, including Notch1, SRY-box transcription factor 9 (SOX9), SOX2, NANOG and octamer-binding transcription factor 4 (OCT4), leading to a cancer stem-like cell phenotype [173].

In pancreatic cancer, overexpression of reactive oxygen species (ROS)-detoxifying genes superoxide dismutase 2 (SOD2) and catalase (CAT) and downregulation of gemcitabine-metabolizing enzyme deoxycytidine kinase (DCK) confers cellular chemoresistance through exosome-derived miRNA-155 [174]. Another nc-miRNA associated with cellular stress, lncRNA-RoR, has been reported to act as a mediator of cell-to-cell communication in hepatocellular cancer, which elevates miRNA TGF levels in recipient cells, resulting in chemoresistance [47].

**Table 1 cancers-13-00084-t001:** Roles and mechanisms of microRNAs (miRNAs) and long noncoding (lnc)-miRNAs reported in cancer pathogenesis.

Exosome Components	Cancer Type	Cell Function	Induced Mechanism	Reference
miRNA-202-3p	Chronic lymphoblastic leukemia (CLL)	Inhibits cancer initiation	Discarded by tumor cells in extracellular vesicles (EVs)	[61]
miRNA-19b miRNA-20a	Acute myeloid leukemia (AML)	Multidrug resistance	Transfer of multidrug resistance protein-1 (MRP-1)	[172]
miRNA-126	Chronic myelogenous leukemia (CML)	Leukemic stem cell quiescence and leukemia growth	Not defined	[109]
LncRNA-MONC miRNA 100HG	Acute megacaryobastic leukemia	Tumor growth	Oncogenes	[71]
miRNA-103a	Lung cancer	Cancer progression and angiogenesis	Decreased phosphatase and tensin homolog (PTEN) and M2 polarization of protumoral macrophages	[64]
miRNA-21	Lung cancer	Modulates immunity, promotes angiogenesis	Increase in ligands of long terminal repeats (LTRs) in immune cells, vascular endothelial growth factor (VEGF) levels	[28,55]
				[75,132]
miRNA-21	Ovarian cancer	Suppresses apoptosis (drug resistance)	Binding to apoptotic protease activating factor 1 (APAF1)	[175]
miRNA-21	Glioblastoma	Priming tumor microenvironment	Microglial cell reprograming	[176]
miRNA-21	Esophageal squamous cell carcinoma	Cancer cell migration and invasion	Activator of cancer-associated fibroblasts (CAFs), cancer cell migration	[177]
miRNA-21	Breast cancer	Tumor progression	Cancer cell stemness and epithelial-mesenchymal transition (EMT), induction of proinflammatory and pro-tumorigenic monocyte profile	[62,178]
miRNA-9	Breast cancer	Promotes angiogenesis tumor metastasis	Janus kinase-signal transducer and activator of transcription (JAK-STAT) activation Induction of CAFs	[78,179]
miRNA-9-5p miRNA-195-5p miRNA-203a-3p	Breast cancer	Stimulate cancer stem-like line phenotype	Transcription factor one cut homeobox 2 (ONECUT2)	[173]
miRNA-939 miRNA-105	Breast cancer	Destruction of endothelial barrier	Downregulation of vascular endothelial (VE)-cadherin, tight junction protein Zonula occludens-1 (ZO-1)	[113,133]
miRNA-105	Breast cancer	Tumor growth	CAF mediation of metabolic reprograming	[180]
miRNA-10b	Breast cancer	Cell invasion	Suppression target genes homeobox D10 (HOXD10) and Kruppel-like factor 4 (KLF4)	[114]
miRNA-200 miRNA-122	Breast cancer	Promote metastasis	Mesenchymal-to-epithelial transition (MET) regulation process, glucose metabolism reprogramming	[108,116]
miRNA-181c	Breast cancer and metastatic brain cancer	BBB destruction Brain metastasis	Downregulation of gene phosphoinositide dependent protein kinase 1 (PDPK1)	[115]
miRNA-221/222	Breast cancer	Drug resistance	Reduction in expression of target genes P27 and ERa	[170]
miRNA-222/223	Breast cancer	Breast cancer cell dormancy in bone marrow and drug resistance	Not defined	[148]
miRNA-143	Breast cancer	Promotion of cancer cell stemness and EMT phenotype	Not defined	[62]
miRNA-143	Prostate cancer	Inhibition of cell growth	Induce death signaling between normal and cancer cells	[45]
miRNA-203 miRNA-212-3p	Pancreatic cancer	Immune dysfunction, immune tolerance	Toll-like receptor 4 (TLR4) regulation, downregulation of regulatory factor X-associated protein (RFXAP) expression	[122,133]
miRNA-155	Pancreatic cancer	Chemoresistance Tumor invasion and progression	Promotion of reactive oxygen species (ROS) detoxification Reprograming of normal fibroblasts into CAFs via tumor protein P53 inducible nuclear protein 1 (TP53INP1)	[110,174]
miRNA-21/155	Neuroblastoma	Resistance to chemotherapy	Crosstalk with miRNA-21 Activation of toll-like receptor 8/nuclear factor Kappa B (TLR8/NFKB) and telomeric repeat binding factor 1 (TERF1) axis	[171]
LncRNA DANCR	Glioma	Tumor progression and malignancy	Sponging miRNA-33a-5p	[73]
miRNA-146a	Multiple myeloma	Tumor cell growth	Increased cytokine and chemokine secretion	[59]
miRNA-24-3p	Nansopharyngeal carcinoma (NPC)	Tumor growth	Target fibroblast growth factor 11 (FGF11) to suppress T cells	[181]
Let-7 family	Gastric cancer	Suppression of cancer initiation	Not defined	[182]
miRNA-15b-3a	Gastric cancer	Tumor progression	Restraining dynein light chain Tctex-type 1 (DYNLT1)/caspase-3/Caspase-9 signaling pathway	[183]
miRNA-203	Colorectal cancer	Metastasis	Differentiation of monocytes to M2 tumor-associated macrophages	[63]
miRNA-210	Hepatocellular carcinoma	Angiogenesis	Inhibition of Mothers against decapentaplegic homolog 4 (SMAD4) and Signal transducer and activator of transcription 6 (STAT6) secretion by endothelial cells	[109]
miRNA-103	Hepatocellular carcinoma	Increase in vascular permeability	Inhibition of VE-cadherin, P120-catenin and zonula occludens 1 expression	[184]
LncRNA-RoR	Hepatocellular cancer	Tumor growth	Sponge miRNA-145 and promote hypoxia-inducible factor (HIF)	[72]
LncRNA-HOTAIR	Urotheral bladder cancer	Tumor initiation and progression	Sponge miRNA-205	[68]
LncRNA-MALATI	Cervical cancer	Tumor invasion	Modulation of epithelial-to-mesenchymal transition (EMT)	[69,70]

### 2.2. Neurodegenerative Disease

In the central nervous system (CNS), close interactions between neurons, microglia, astrocytes and oligodendrocytes facilitate nerve homeostasis, cellular communication and signal transduction by secreting exosomes, which, however, also leads to the transfer of abnormal mediators [185]. These exosomes, which are released into the extracellular microenvironment, have, in recent years, led to increased interest in the pathophysiology of neurodegenerative diseases associated with aging and increasing life expectancy. Alzheimer’s disease (AD), frontotemporal dementia, Parkinson’s disease (PD), Huntington’s disease (HD), multiple sclerosis (MS) and amyotrophic lateral sclerosis (ALS) have been the subject of intense study focused on different aspects of these diseases, including their physiology, etiopathology, diagnosis and biomarkers, as well as emerging treatments [16,186]. These pathologies are characterized by protein aggregates and the formation of inclusion bodies in specific sites in the brain due to neuronal cell death. The impairment of the quality control mechanisms of these proteins resulting from age-related external stress induces the transmission of these aggregates to other aggregate-free cells in the brain [186]. Recently, exosomes have been identified as potential new biomarkers of great interest in synaptic transmission and nerve regeneration. Additionally, some evidence shows that they are involved in pathogenesis and could play a role in the advanced treatment of neurodegenerative diseases. These exosomes, which act as key mediators in intercellular communication, have recently been observed to be involved in age-related neurodegenerative diseases, leading to cognitive impairment due to their ability to transmigrate the blood–brain barrier (BBB) and to transfer pathological protein aggregates such as amyloidβ (Aβ), tau and α-synuclein proteins to distant brain cells [187]. Cancer cell-derived exosomes can reach the CNS by destroying the BBB and transferring to neural cells. miRNA-181-c has been shown to activate actin mislocalization, enabling exosomes to be transferred to the periphery of the CNS [115]. There is also evidence that exosomes have the ability to cross the BBB in the opposite direction. Hematopoietic cells are reported to transfer their exosomes to Purkinje cells in the brain, leading to a modification in gene expression via the inflammatory pathway [188]. Moreover, exosomes are involved in nerve injuries associated with infectious agents. Prion proteins might be taken up in the infected cells and then delivered to target cells through membrane fusion after secretion in the extracellular fluid [21], suggesting that they play a role in spreading the infectious disease in the brain.

AD is the first common neurogenerative disease in which affected neurons probably secrete tau protein in the exosomes released, thus contributing to the spread and progression of tauopathy due to tau protein hyperphosphorylation [189]. Wang et al. have demonstrated that neuron depolarization leads to the release of exosome-derived tau, whose trans-synaptic transmission is confirmed by its trans-neuronal and microglial transfer [190,191]. Exosomes effectively spread within interconnected neurons and transfer Aβ and tau proteins through an endosomal pathway and axonal transport [192]. The exosomal hyperphosphorylated tau (p-tau) protein and the extracellular senile plaque containing the Aβ peptide result in neuron degeneration and the secretion of proinflammatory cytokines by microglia and astrocytes, thus altering the BBB and causing AD [193]. Rajendran et al. reported that exosome-derived p-tau protein concentrations increase significantly in the mild/severe stages (Braak stages 3–6) of AD, as compared to patients in the early stages (Braak stages 0–2), suggesting that exosomes play a crucial role in the abnormal processing of tau in the cerebrospinal fluid (CSF) in early onset AD [194,195]. On the other hand, Aβ is transported by exosomes to be degraded by lysosomes in normal settings, and the disruption of this clearance could lead to their accumulation in exosomes and AD spread [196]. Similarly, this lysosomal dysfunction has been observed in relation to exosomal α-synuclein release and transmission [197]. Disruption of the secretory pathway of neurons is another pathological mechanism leading to AD, in which the neuroprotective signal peptide sequence targeted by cystatin C is downregulated in exosomes [198]. The soluble amyloid protein precursor (APP) is thus decreased and associated with the involvement of Aβ aggregates [199]. Exosomes from activated astrocytes have also been observed in the pathogenesis of AD by targeting the inflammatory and proapoptotic pathways [200,201]. Astrocytic damage is related to Aβ senile plaques through the activation of prostate apoptosis response 4 (PAR-4) [202,203], while the exosome secretion of PAR-4/ceramide results in neuroprotective astrocyte apoptosis and AD induction [204].

The neurons are likely to modulate myelin biogenesis by regulating the secretion of oligodendroglia-derived exosomes, whereby myelin sheaths are slowed down during CNS development [205]. These exosomes contain myelin proteins and RNAs involved in promoting myelination [206,207]. Their impact is not restricted to a positive effect on myelination through an increase in neuron resistance to stress and their enhanced growth but might also be involved in repairing damaged myelin sheaths [101].

In an immunological setting, exosomes from astrocytes, microglia, platelets, leukocytes and endothelial cells have been demonstrated to secrete metalloproteinase (MMP)-14 and caspase 1 following stimulation by proinflammatory cytokines in MS. These enzymes facilitate lymphocyte and myeloid cell transmigration to CNS by inducing the disintegration of the BBB [208,209]. In addition, endothelial-derived exosomes transfer the ICAM-1 receptor for integrin Mac-1 to monocytes, thus increasing their transmigration through the barrier [210]. Furthermore, activated T lymphocytes are involved in this immunological cascade by releasing exosomes containing larger amounts of chemokine CCL5, which facilitates their adhesion to brain microvessel endothelium cells [211]. This suggests that exosome generation by the neural and immune cell network is of great importance in MS pathogenesis.

Exosome cargo is also transferred outside the CNS. In MS, serum-derived exosomes have been found to contain three myelin proteins: the myelin basic protein, the proteolipid protein and the myelin oligodendrocyte glycoprotein (MOG). Some evidence indicates that MOG content is strongly associated with MS, which modulates anti-myelin immune reactions in both relapsing-remitting MS (RRMS) and secondary progressive MS (SPMS) patients [212]. Significant sphingomyelinase enzymatic activity has recently been found in MS patient-derived exosomes, resulting in decreasing levels of different sphingomyelins in their CSF, which is associated with axonal damage and neuronal dysfunction [213].

Dopaminergic neuron degeneration in substantia nigra, the formation of intracytoplasmic Lewy bodies in other surviving neurons and the abnormal accumulation of α-synuclein are related to the occurrence of PD [214,215]. In addition, α-synucleins control synaptic transmission and vesicle release [216], where Lewy bodies indicate pathological α-synuclein aggregation in neurons and glial cells [217], which propagate according to a prion-like pattern [218]. Some evidence indicates that exosomes are involved in PD by transporting α-synucleins to lysosomes for degradation, which might then be accumulated and released into the intercellular space, resulting in cytotoxicity [219,220]. The coaggregation of α-synuclein with Aβ and the protein tau has also been reported, thus accelerating the neuropathology and cognitive decline [221,222].

Although protein aggregation is a major cause of neurodegenerative disease, exosome-derived miRNAs play a key role in controlling protein levels by regulating their mRNAs [223,224]. Differential miRNA expression is closely associated with AD, PD, ALS, MS and HD [225,226,227,228,229,230,231]. In MS, different miRNAs have been identified in serum-derived exosomes, whose signatures appear to be indicative of disease subtypes. MiRNA-15b-5p, miRNA-451a, miRNA-30b-5p and miRNA-342-3p have been identified in RRMS patients, while miRNA-127-3p, miRNA-370-3p, miRNA-409-3p and miRNA-432-5p have been found in SPMS patients [232]. Given the T-cell-mediated autoimmune nature of MS, various studies have reported the involvement of miRNAs in CNS immunomodulation. Exosomal miRNA let-7i was found to increase in MS patients and to suppress T-reg cell induction by targeting insulin-like growth factor 1 receptor (IGF1R) and TGF-β receptor 1 (TGF-β R1), leading to autoimmune modulation [233]. Exosomal miRNA let-7 can also activate TLR7 in neuronal cells and trigger inflammation, causing neuronal death [234,235]. On the other hand, Winkler et al. have suggested that neurons activate TLR7 proteins present in endosomes and the uptake of exosomes containing miRNA let-7, thus inducing cell degeneration [236]. In the CNS, TLRs are widely expressed in different cell types, whose crosstalk with miRNAs is associated with immune damage, causing inflammation and neurodegenerative diseases. Additionally, the pathogenesis of MS is related to an increase in miRNA-326 secretion from T-cell-derived exosomes in RRMSs, thus targeting TH17 differentiation and maturation [237].

In AD, Aβ and the hyperphosphorylated tau protein are individually regulated by the APP gene. Increased APP activity results in higher Aβ levels, which negatively impacts synaptic function and neuron degeneration [238]. Various studies have reported that miRNA-16; miRNA-101; miRNA-193b; miRNA-200b and the miRNA-20a family (miRNA-20a, -106b and -17-5p) downregulate APP expression [239,240,241]. On the other hand, the post-transcriptional protein tau is targeted by miRNA-34a by combining with the 3′-untranslated region (UTR) of microtubule-associated protein tau (MAPT), which inhibits its endogenous expression and leads to AD neuron degeneration [242,243].

The α-synuclein protein characterizing PD pathogenesis has been found to be overexpressed, with a recent study reporting that the α-synuclein gene (SNCA) combines its 3′-UTR mRNA with miRNA-7, resulting in the inhibition of transcription and protein expression. In PD, given the decrease in miRNA-7 expression, α-synuclein was found to be toxic to dopamine neurons [244,245]. In addition, the blood plasma of patients is enriched in miRNA-4639-5p as a result of the post-transcriptional downregulation of the DJ-1 gene, given that the decrease in DJ-1 protein levels causes severe oxidative stress and neuron death [230].

## 3. Exosome Composition

Exosomes contain numerous molecules, including proteins, lipids, metabolites, mRNA and microRNA [246], as well as genomic and mitochondrial DNA [247,248]. Other forms of RNA, including transfer, ribosomal, small nucleolar and long noncoding RNA (lncRNA), have also been identified [249] (Figure 1). These can be transferred from host to recipient cells in order to regulate cellular functions [250,251,252]. In addition, the ExoCarta, EVpedia and Vesiclepedia exosome databases provide detailed information regarding the molecular content of exosomes [253]. The composition of exosomes is a tightly regulated process that is influenced by environmental factors such as cell activation and stress conditions [254]. Exosomes secreted by the same cells are expected to have a similar protein, lipid and nucleic acid composition. However, the molecular composition of exosomes has recently been shown to be non-cell type-dependent and differs even when the exosomes originate from the same parental cells [255,256,257]. On the other hand, some cargos are common to exosomes of different origins [258]. Novel methods and technologies, including high-resolution flow cytometry [259], laser tweezer Raman spectroscopy (LTRS) [257], ultracentrifugation [260] and immunocapturing [261], have recently been developed in order to differentiate features of exosomes such as exosomal heterogeneity [262].

### 3.1. Nucleic Acids

Exosomes contain nucleic acids, including messenger RNA (mRNA), microRNA (miRNA) and other noncoding RNAs, which can be transferred between cells and possibly regulate gene expression in recipient cells [263]. Exosomes released from cancer patients have been found to contain fragments of single-stranded DNA and double-stranded genomic DNA, including all chromosomes [264,265]. These vesicles also excrete harmful DNA from cells in order to maintain cellular homeostasis [266]. Exosomal RNA content is a subset of cellular RNA and, in some cases, may differ significantly from that of its parent cell. However, other RNAs are ubiquitous among all types of exosomes regardless of their cell of origin due to their specific targeting in multivesicular bodies (MVBs) during biogenesis [267], indicating that specific RNAs are actively sorted into exosomes. In addition, miRNA packaging in EVs is different from that of the parent cell and is particularly influenced by external stimuli. As exosomal miRNAs play a prominent role in disease progression, induce angiogenesis and facilitate metastasis in cancers [112,268], they can be used as potential noninvasive biomarkers of disease states [269,270].

Koppers-Lalic and colleagues have suggested that post-transcriptional modifications, notably 3’-end adenylation and uridylation, have opposite effects that may contribute, at least in part, to directing ncRNA sorting towards EVs, given the overrepresentation of 3′-end-adenylated miRNAs and 3′-end-uridylated miRNAs in human B cells and their secreted exosomes, respectively [271]. Dicer and Ago2, key components of miRNA processing, have been found to be functionally present in exosomes [272]. A tetranucleotide sequence is also present in miRNAs that controls their localization in exosomes. In fact, the protein heterogeneous nuclear ribonucleoprotein A2B1 (hnRNPA2B1) specifically binds exosomal miRNAs through the recognition of this sequence and controls their loading into exosomes [273]. Similarly, the synaptotagmin-binding cytoplasmic RNA-interacting protein (SYNCRIP) can control miRNA sorting in exosomes. This protein binds directly to miRNAs enriched in exosomes that share a similar sequence or hEXO motif. This motif, whose introduction into a poorly exported miRNA improves its exosomal loading, can regulate miRNA localization [274].

Exosomes produced by endothelial cells promote angiogenesis in vivo in a small RNA-dependent manner. Exosomes produced by human breast cancer cell lines MDA-MB-231 and MDA-MB-436 contain various classes of RNA, such as small nucleolar RNAs (snoRNAs), ribosomal RNAs (rRNAs), transfer RNAs (tRNAs), microRNAs (miRNAs) and yRNAs, with the major class of RNA being fragmented rRNAs, particularly rRNA subunits 28S and 18S [275]. On the other hand, tRNAs are the most common RNA species found in exosomes derived from human adipose- and bone marrow-derived mesenchymal stem cells (MSCs). More than 50% of total small RNAs are tRNAs in adipose-derived exosomes (ASC), while tRNAs account for 23–25% of the total small RNA content in bone marrow (BMSC) exosomes [276]. Similarly, exosomes isolated from urine contain high concentrations of rRNAs (40–60%) and tRNAs (20–50%), followed by mRNAs (5–15%) and miRNAs (5–10%), while serum-derived exosomes are enriched with miRNAs (30–75%), mRNAs (10–20%) and tRNAs (20–30%) [277]. As tRNAs can bind to argonaut proteins and recognize mRNA targets similar to miRNAs, tRNAs may play a major role in RNA silencing [278]. Furthermore, vault RNAs (vRNAs) have been reported to play an important role by mediating the drug-resistant phenotype of malignant cells, suggesting that vRNAs may be involved in the sequestration of chemotherapeutic compounds. On this basis, mitoxantrone has the ability to bind to vRNAs, which potentially sequesters the drug and prevents it from reaching the target site [279].

### 3.2. Proteins

Exosome protein contents have been well-identified using a wide variety of proteomic techniques. High-throughput proteomic analyses have revealed the presence of proteins involved in cell structure, motility and adhesion, such as actins, myosin, radixin, tubulins, integrins, and cell surface receptors, including epidermal growth factor receptors (EGFRs), platelet-derived growth factor receptor beta (PDGFRB) proteins and plasminogen activator urokinase receptors (PLAURs), as well as signaling proteins, transcription factors and metabolic enzymes [280,281]. In addition, ExoCarta has indicated the presence of over 4000 proteins in exosomes. Exosomal protein composition can vary between different cell types and under different culture conditions. Ingenuity pathway analysis (IPA) has identified the presence of 157 proteins in placenta mesenchymal stem cell (PlaMsc)-derived exosomes exposed to 1% O_2_. On the other hand, 34 and 37 individual proteins were found to be present in PlaMSC-3%O_2_ and PlaMSC-8%O_2_ exosomes, respectively. More proteins associated with vascular endothelial growth factor (VEGF), actin cytoskeleton, growth hormone and clathrin-mediated endocytosis signaling in exosomes have been reported to be isolated from pMSC exposed to 1% O_2_ as compared to 3% or 8% O_2_, possibly leading to an increase in the exosome uptake of target cells [282]. As characterized by matrix-assisted laser desorption ionization time-of-flight (MALDI-TOF) analysis, MHC-I, together with heat shock proteins HSC70 and HSP90, annexins, PV-1 and developmental endothelial locus-1 (DEL-1), were found to be present in exosomes derived from human mesothelioma cells [283].

Certain molecular markers commonly found in exosomes are essential for the overall biological and pharmacological effects of exosomes. Heat shock proteins HSP70 and HSP90 are molecular chaperones, and tumor susceptibility gene 101 (TSG101) is involved in multivesicular body (MVB) biogenesis. Moreover, tetraspanin and integrin proteins such as CD63, CD9, CD81 and CD82 are pivotal for cell targeting and adhesion, while Rab GTPases, annexins and flotillin are involved in membrane transport and fusion [284]. Different α and β chains of integrins (α4β1, αMβ2, αLβ2 and β2); A33 antigen and P-selectin; ICAM1/CD54 and cell-surface peptidases CD26 and CD13 are also present in exosomes [285]. Interestingly, given their competition with membrane MHC-II for T-cell receptor binding on CD4^+^ T cells, soluble MHC-II (sMHC-II) proteins play a prominent role in immune response suppression and the maintenance of tolerance [286].

As the protein composition of exosomes is not identical to that of the parent cell, there are two major protein sorting pathways: the dependent and independent endosomal sorting complexes for transport (ESCRT). ESCRT are composed of four multimeric complexes, ESCRT-0 to ESCRT-III. Baietti and colleagues showed that cytoplasmic adaptor syntenin interacts directly with ALIX, which, in turn, binds to ESCRT-III and is involved in the sorting of syndecan membrane proteins in exosomes [287]. On the other hand, other studies have indicated that proteins can also be packaged into MVBs without the involvement of ESCRT or ubiquitin. Intraluminal vesicle (ILV) formation and melanosomal protein (Pmel17) sorting continue following the disruption of the Hrs/ESCRT function, suggesting that Pmel may be sorted into ILVs independently of Hrs/ESCRT machinery [288]. In addition, the features of protein Sna3p enable its selective inclusion in invaginating vesicles independently of ubiquitin [289]. Intriguingly, Lin et al. found that many ribosomal proteins are secreted by exosomes that are derived from embryonic fibroblasts in sirtuin 6 knockout mice, indicating that SIRT6 affects the sorting of many proteins to exosomes [290].

Le Pecq and colleagues showed that dendritic cell-derived exosomes (dexosomes) induce strong antitumor activity by displaying antigens to CD8^+^ and CD4^+^ T cells. In addition, this form of immunotherapy is well-tolerated in patients with advanced non-small cell lung cancer (NSCLC), thus rendering dexosomes a viable acellular alternative to dendritic cells (DC) for use in cancer vaccinations in preclinical and clinical studies [291,292]. Some highly potent proteins in MSC-derived exosomes have the potential to improve cardiac function after myocardial infarction (MI), including growth factors such as fibroblast growth factor 1 (FGF1) and neuregulin-1 (NRG1), involved in cardiac development and regeneration in an MI rat model [293]. In addition, cardiac-specific human fibroblast growth factor 1 (FGF-1) is also associated with enhanced postischemic hemodynamic recovery and the attenuation of reperfusion-induced myocardial cell necrosis during ischemia reperfusion (IR) [294]. Macrophage colony-stimulating factor (M-CSF) increases vascular endothelial growth factor (VEGF) production from cardiomyocytes, protects cardiomyocytes and myotubes from cell death and enhances cardiac function after ischemic injury [295]. Hill et al. demonstrated that glial growth factor 2 (GGF2) improves cardiac function in rats with MI-induced systolic dysfunction [296]. Similarly, chronic leukemia inhibitory factor (LIF) treatment has a positive effect on systolic heart function, suggesting that LIF may have a therapeutic role in preventing or repairing myocardium injury [297].

### 3.3. Lipids

The effects of exosomes are not only mediated by their nucleic acid and protein content, but exosomal lipids, in particular, can also modulate their bioactivity and vesicle stability. Understanding the biological and pharmacological effects of exosomal lipids can improve our knowledge of exosome biogenesis and will help to develop efficient exosome-based therapeutics [262].

Exosomes are a heterogeneous population of extracellular vesicles (EVs) with different surface-expressed molecular patterns, thus providing an additional tool for their identification. The lipid composition of exosomes, which accounts for their unique rigidity, differs from that of the parent cell’s plasma membrane, partly because exosomes also contain lipids from the Golgi apparatus. These vesicles are also rich in cholesterol, ceramide and other sphingolipids, as well as phosphoglycerides with long saturated fatty acyl chains [298]. In this regard, B-cell-derived exosomes are rich in ceramides [299], whose role in the budding of exosome vesicles into MVBs has also been reported [298]. On the other hand, exosomes secreted from oligodendrocytes are highly rich in phosphatidylcholine (40%), phosphatidylserine (25%) and phosphatidylethanolamine (20%) but contain only 2.2% cholesterol [300].

Exosomes from mast and dendritic cells have increased levels of phosphatidylethanolamines, which have a higher rate of flipping between the two leaflets of the exosome bilayer than in cellular membranes [301]. Interestingly, exosomes are able to deliver prostaglandins to the target cells and carry prostaglandins bound to the exosomal membrane with potentially enhanced biological activity rather than the soluble form of prostaglandins [302]. Recent studies have shown that exosomes may affect the lipid composition of recipient cells, specifically cholesterol and sphingomyelin, thus modulating cell homeostasis [303]. Beloribi-Djefaflia and colleagues suggested that exosomal lipids contribute to tumor progression and drug resistance in Mia-PaCa-2 cells [304]. Finally, ceramide-enriched exosomes have been shown to induce astrocyte apoptosis, potentially contributing to the progression of Alzheimer’s disease [204].

## 4. Applications of Exosomes in Biomedicine

### 4.1. Exosomes as Biomarkers

Exosomes are now regarded as new players in regenerative medicine due to their therapeutic capacity and their potential as noninvasive biomarkers for early diagnosis; the evaluation of treatment efficacy and monitoring of the progression of cancer, neurodegenerative, metabolic and infectious diseases [5,305]. They offer a simple method for the molecular analysis of biofluids that reduces invasive surgery and promotes more precise medical interventions. Several clinical trials have been launched for both early screening and accurate diagnosis to reduce mortality rates and to increase recovery rates (Table 2). The molecular content of exosomes reflects the origin and pathophysiological conditions of releasing cells, suggesting that the analysis of exosomal markers is a highly specific and sensitive method that could potentially replace invasive biopsies. In addition, their small volume, specific biological information, strong permeability through body tissue barriers, abundance and long half-lives in all biological fluids make these biomarkers highly attractive targets for clinical diagnostic applications. In addition to nucleic acids, exosomal proteins have been found to be potential biomarkers for a variety of pathologies, including cancer, as well as a number of noncancer diseases in different organs, such the central nervous system [195,197], the kidneys [306,307], liver [308] and lungs [309].

#### 4.1.1. Exosomes for Cancer Diagnosis

Several types of cancer have long been known to shed exosomes into the blood. Fortunately, recent technological advances have enabled the capture and analysis of these cancer-derived exosomes to be improved upon, making them valuable diagnostic tools. RNAs, including mRNAs, lncRNAs, circular RNAs (circRNAs) and miRNAs, DNA, proteins and lipids, have been extensively used as cancer biomarkers (Figure 3).

DNA. Exosomes produced by several cancer types have been reported to contain DNA. These vesicles carry either long double-stranded DNA fragments [310] or single-stranded DNA [264]. Some studies have revealed the presence of double-stranded DNA in exosomes secreted by human carcinoma and murine melanoma, suggesting its potential use in the early clinical detection of cancer [248]. Similarly, Kahlert and coworkers detected the predominance of double-stranded DNA in pancreatic cancer-derived exosomes, as well as similar genomic mutations among exosomes and parental cancer cells [265]. On the other hand, Balaj et al. identified single-stranded DNA in medulloblastoma-derived exosomes, thus illustrating its promising potential use in cancer diagnosis and therapy [264].

Messenger RNAs (mRNAs). Increased levels of epidermal growth factor receptor variant type III (EGFRvIII) mRNA have been detected in the serum exosomes of glioblastoma patients, suggesting its use as a new glioblastoma diagnosis method instead of surgery [270]. Exosome Diagnostics, Inc. (Waltham, MA, USA) have developed methods for detecting one or more biomarkers in urine microvesicles in order to aid the diagnosis, monitoring and treatment of diseases such as cancer, especially prostate gland-related pathologies. Biomarkers, which are mRNAs of one or more isoforms of a large group of genes, facilitate the detection of prostate cancer by determining the fusion between SLC45A3 and BRAF genes in urinary microvesicles [311]. Recently, Dong and coworkers found that exosomal serum membrane type 1-matrix metalloproteinase (MT1-MMP) mRNA increases significantly in gastric cancer (GC) patients, which correlates with the tumor, lymph node and metastasis (TNM) stage and lymphatic metastasis. These findings indicate that exosomal MT1-MMP mRNA can be utilized as a biomarker for GC diagnosis and early treatment [312]. Similarly, exosomal heterogeneous nuclear ribonucleoprotein H1 (hnRNPH1) mRNA levels, which are remarkably higher in hepatocellular carcinoma (HCC) patients than in other groups, are associated with the Child-Pugh and TNM stage classification, portal vein tumor emboli and lymph node metastasis. This confirms that exosomal serum hnRNPH1 mRNA could be an effective marker of HCC [313]. Esophageal cancer-related gene-4 (Ecrg4) has been shown to be a tumor suppressor in several studies. Mao and colleagues have reported that serum exosomes contain higher levels of ECRG4 mRNA in healthy individuals than in their cancer counterparts, thus showing that exosomal ECRG4 mRNA can be used for cancer detection [314].

MicroRNAs (miRNAs) are small noncoding, double-stranded RNA molecules that degrade complementary mRNA sequences in target cells in order to inhibit protein translation. These molecules are reported to be abnormally expressed in several types of cancer, suggesting their role in the pathogenesis of human cancer [315]. Eight miRNAs, previously shown to be diagnostic markers of ovarian cancer, have been reported to be present at similar levels in biopsy specimens of ovarian cancer and circulating exosomes isolated from the same ovarian cancer patients [316]. With respect to lung tumors, Rabinowits and coworkers found similar miRNA patterns in plasma exosomes and tumor biopsies from lung adenocarcinoma patients. However, miRNA levels in lung cancer patients and control subjects differed significantly, indicating that circulating exosomal miRNA could be useful for lung adenocarcinoma screening tests [269]. Hepatocellular carcinoma (HCC) is a primary liver malignancy and a leading cause of cancer-related mortality worldwide. Exosomal miRNA-210 secreted by hepatocellular carcinoma cells is reported to promote angiogenesis by targeting SMAD4 and STAT6 in endothelial cells. Therefore, exosomal miRNA-210 could be used as a therapeutic target in anti-HCC therapy [109]. In this regard, circulating miRNAs in serum exosomes have potential as novel biomarkers for predicting hepatocellular carcinoma recurrence following liver transplantation [317]. In addition, Takeshita and colleagues reported that the sensitivity and specificity of serum miRNA-1246 in an esophageal squamous cell cancer (ESCC) diagnosis are 71.3% and 73.9%, respectively. Serum miRNA-1246, which closely correlates with the tumor, lymph node and metastasis (TNM) stage, has been shown to be a strong independent risk indicator of poor survival rates. Intriguingly, miRNA-1246 levels were found to be elevated in serum exosomes from ESCC patients but not in ESCC tissue samples, suggesting that exosomal serum miRNA-1246 could be a valuable diagnostic and prognostic biomarker of ESCC [318]. Circulating exosomal miRNA-17-5p and miRNA-92a-3p were found to be upregulated in colorectal cancer (CRC) patients. Their expression levels correlated closely with metastasis and chemotherapy resistance [319]. Moreover, exosomal miRNA-320d has been identified as a promising blood-based biomarker for distinguishing metastatic from nonmetastatic diseases in the serum of CRC patients. Therefore, these noninvasive biomarkers may have great potential to predict the clinical behavior of CRC and to monitor tumor metastasis [320,321]. Mitchell et al. reported that circulating miRNA-141 levels are strong diagnostic markers of prostate cancer [322]. Furthermore, exosomal serum miRNA-141 and miRNA-375 have been found to correlate with tumor progression in prostate cancer [323]. The enrichment of the let-7 miRNA family in exosomes from AZ-P7a cells may reflect their oncogenic characteristics, including tumorigenesis and metastasis, suggesting that AZ-P7a cells release let-7 miRNAs via exosomes into the extracellular environment to maintain their oncogenesis [182].

Long noncoding RNAs (lncRNAs). Exosomes also contain lncRNAs, now characterized as potential diagnostic and prognostic biomarkers for a wide range of pathologies. These functional RNAs, which are longer than 200 nucleotides, do not code for proteins but, rather, bind to a variety of nucleic acids and proteins as a means to regulate gene expression at the transcriptional and/or translational level. Colon cancer-associated transcript 2 (CCAT2), a novel lncRNA transcript encompassing the rs6983267 SNP, is significantly upregulated in CRC tissues as compared to adjacent noncancerous tissues. The higher expression levels of CCAT2 are associated with a greater depth of local invasion, positive lymph node metastasis and advanced TNM stage [324]. Moreover, exosomal lncRNA and miRNA-217 are differentially expressed in the serum of colorectal carcinoma patients and correlate with tumor classifications (T3/T4), advanced clinical stages (III/IV) and lymph node or distant metastasis [325]. LncRNA 91H is known to play a prominent role in tumor development by enhancing tumor cell migration and invasion through the modification of heterogeneous nuclear ribonucleoprotein K (HNRNPK) protein expression. In addition, CRC patients with high lncRNA 91H expression demonstrate a higher risk of tumor recurrence and metastasis [326]. Interestingly, exosomes from healthy donors carry a significant amount of HOTTIP (HOXA distal transcript antisense RNA) transcripts in comparison to CRC patients, with a significant statistical correlation between low exosomal HOTTIP levels and poor overall survival rates. Therefore, lncRNA *HOTTIP* could be a viable biomarker for CRC patients to predict the postsurgical survival time [327]. Exosomal serum lncRNA HOTAIR (Hox transcript antisense intergenic RNA) and miRNA-21 expression levels were higher in patients with lymph node metastasis than those without. In addition, exosomal HOTAIR and miRNA-21 achieved a sensitivity and specificity of 94.2% and 73.5%, respectively, in differentiating the malignant from benign laryngeal disease, suggesting that the combined evaluation of their serum expression levels may be a valuable biomarker of laryngeal squamous cell carcinoma [328].

Proteins. Exosomal protein signatures have also been used as potent alternative diagnostic markers of cancer. The epidermal growth factor receptor (EGFR) localized to exosome membranes has been found to be a possible marker for lung cancer diagnosis [329]. In this regard, Jakobsen and coworkers reported that the EGFR is highly expressed on the exosomal surface by analyzing the extracellular vesicles secreted by lung cancer cells [330], indicating that the EGFR is a promising biomarker for diagnosing non-small cell lung cancer (NSCLC). The epidermal growth factor receptor variant type III (EGFRvIII) transcript was detected in serum exosomes from 25 spongioblastoma patients but was not found in serum exosomes from 30 normal control individuals. Therefore, exosomal EGFRvIII may provide diagnostic information for glioblastoma patients [270]. Similarly, Graner et al. reported that brain tumor exosomes can escape from the blood–brain barrier, with potential systemic and distal signaling and immune consequences, and that serum exosomes from brain tumor patients possess EGFR, EGFRvIII and TGF-beta [331]. A microfluidic chip was used to analyze exosomal protein types in the blood circulation of spongioblastoma patients. In this regard, Shao and colleagues found that circulating exosomes contain EGFR-VII, EGFR, PDPN and IDH1, which can be used to analyze primary tumor mutations and to indicate drug efficacy [332]. Urinary exosomal proteins have also been investigated as potential biomarkers for prostate and bladder cancers. Nilsson et al. showed that urinary exosomes in prostate cancer patients express prostate-specific antigen (PSA), prostate cancer gene-3 (PCA-3), transmembrane serine protease 2-erythroblast transformation-specific (ETS) transcription factor family member-related gene fusion (TMPRSS2-ERG) and other prostate cancer-related markers, indicating the potential for the diagnosis and monitoring of cancer patients [333]. In this respect, Chen and colleagues found that 24 urinary exosomal proteins presented at significantly different levels in hernia (control) and bladder cancer patients. In particular, they revealed the strong association of TACSTD2 with bladder cancer and the potential of human urinary exosomes in noninvasive cancer diagnosis [334]. CD24, found in the MVB cytoplasm, is released into the extracellular environment via exosomes and is associated with the poor prognosis of ovarian carcinomas [335]. Logozzi and colleagues found that plasma CD63+ exosome levels are significantly higher in melanoma patients as compared to healthy control individuals [336]. This team recently showed that plasmatic exosomes from prostate cancer patients overexpress carbonic anhydrase IX (CA IX), as well as CA IX-related activity. In addition, CA IX expression correlated with intraluminal acidity in the plasmatic exosomes of these cancer patients [337]. The acidic microenvironment was reported to induce an upregulation of both the expression and activity of CA IX in cancer-derived exosomes, along with an increase in their production levels [338]. Finally, leucine-rich alpha-2-glycoprotein 1 (LRG1) expression levels were found to be higher in the urinary exosomes and lung tissue of NSCLC patients as compared to healthy individuals, indicating that LRG1 may be a candidate biomarker for noninvasive NSCLC diagnosis [309].

Lipids. Exosome lipidomics show great potential for the identification of suitable markers for cancer diagnosis. Recently, using an untargeted high-resolution mass spectrometry approach, our research group identified similarities between structural lipids differentially expressed in cancer stem cell (CSC)-derived exosomes and those derived from patients with malignant melanoma (MM) [339]. Our results showed significant metabolomic differences between exosomes derived from MM CSCs and those from differentiated tumor cells and, also, between serum-derived exosomes from patients with MM (MMPs) and those from healthy controls (HCs). We detected metabolites from different lipid classes, such as glycerophosphoglycerols, glycerophosphoserines, triacylglycerols and glycerophosphocholines. Interestingly, we found that PC 16:0/0:0 glycerophosphocholine expression was lower in both CSCs and MMPs in comparison with differentiated tumor cells and HCs, respectively, while lysophospholipid sphingosine 1-phosphate (S1P) levels were found to be lower in serum-derived exosomes from MMP patients than from HCs. These results indicate the importance of structural lipids detected in exosomes as biomarkers in the early detection of cancer and their potential in the determination of aggressiveness and therapeutic monitoring [339].

#### 4.1.2. Use of Exosomes for Molecular Diagnostics of Neurodegenerative Diseases

Recent evidence indicates the potential involvement of exosomes in the nervous system and highlights their role in transcription regulation, neurogenesis and plasticity [340]. Several central nervous system (CNS) cell types, such as neurons and glial cells, are known to communicate intercellularly by releasing EVs. However, these vesicles could also play a role in the development of neurodegenerative diseases. Parkinson’s disease (PD) is a progressive neurodegenerative disorder that mostly affects the motor system. Proteomic profiling was used to differentially identify proteins expressed in serum exosomes from PD patients and healthy individuals [341]. In addition, Fraser and colleagues identified leucine-rich repeat kinase 2 (LRRK2) as a biomarker in urinary exosomes from PD patients that predicts the risk of the development of this disease among LRRK2 mutation carriers [342]. The aggregation of α-synuclein may play an important role in PD pathology. Exosomes have been shown to be able to transfer the α-synuclein protein to neighboring normal cells, thus possibly exacerbating PD pathogenesis [197].

Alzheimer’s disease (AD), another neurodegenerative disorder, is now regarded as the most common casue of dementia. The early detection of exosome-associated tau, which is present in human cerebrospinal fluid (CSF) samples and is phosphorylated at Thr-181 (AT270), would be helpful for AD diagnosis [194]. In this regard, the T-tau, P-tau and neurofilament light (NFL) biomarkers could be used to differentiate effectively between AD patients and healthy subjects [343]. Exosomal lipids could also be used as promising biomarkers for AD diagnosis. In this respect, 10 lipids from plasma were able to predict phenoconversion to AD within a two-to-three-year timeframe with over 90% accuracy [344].

**Figure 3 cancers-13-00084-f003:**
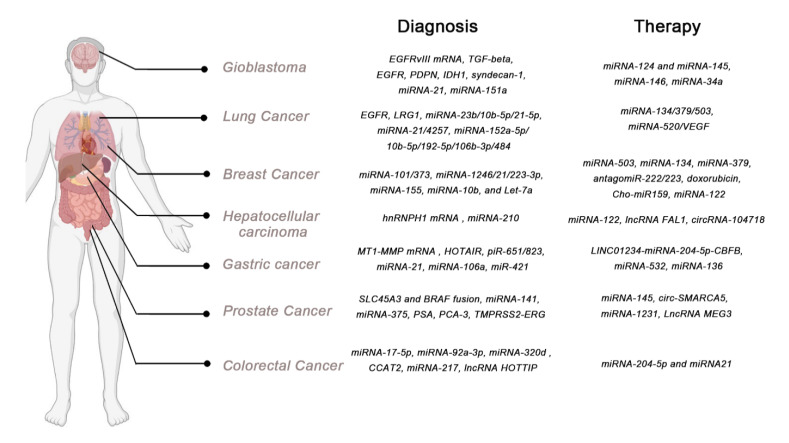
Exosome-associated molecules used for diagnosis and therapy. For instance, epidermal growth factor receptor variant type III (EGFRvIII) is associated with the classical glioblastoma (GBM) subtype [345]. MicroRNA (miRNA)-124 has been reported to enhance the chemosensitivity of GBM cells to temozolomide and to decrease GBM cell migration [346]. In addition, the delivery of miRNA-34a results in the inhibition of GBM cell proliferation, invasion, migration and tumurogenesis both in vivo and in vitro [347]. Lung cancer was also detected using exosomal biomarkers. In this context, Liu et al. found that miRNA-23b/10b-5p/21-5p were good candidates for its diagnosis [348], while Dejima and coworkers considered miRNA-21/4257/451a reliable biomarkers [349,350]. Other miRNAs such as homo sapiens (hsa)-miRNA-320d/320c/320b were suggested as potential biomarkers [351]. On the other hand, exosome miRNA-101/373 serum levels were found to be linked to aggressive breast carcinomas [352]. Other authors recommend miRNA-1246/21/223-3p as potential indicators of breast cancer [353,354]. Therapeutic quantities of doxorubicin (Dox) and cholesterol-modified miRNA 159 (Cho-miRNA-159) were delivered to triple-negative breast cancer (TNBC) cells and exhibited improved anticancer effects [355]. In addition, miRNA-204-5p and miRNA-21 efficiently inhibited cancer cell proliferation and increased chemosensitivity by specifically suppressing their target genes in human colorectal cancer cells [356,357]. Adipose-derived stromal cells (ASCs) were shown to be able to promote prostate cancer cell apoptosis via exosomal miRNA-145 through the caspase-3/7 pathway [358,359].

### 4.2. Use of Exosomes as Therapeutic Agents

In many studies, exosomes have been used as delivery vectors for small-molecule therapeutic agents, as they are capable of traveling from one cell to another and of conveying their cargo in a biologically active form, thus acting as attractive gene and drug delivery vehicles [360]. Cancer cells internalize a significantly larger percentage of exosomes as compared to normal cells. HEK293 and MSC exosomes were therefore effectively used as delivery vectors to transport PLK-1 small interfering RNA (siRNA) to bladder cancer cells in vitro, resulting in the selective gene silencing of PLK1 [361]. In addition, the internalization of exosomes in tumor cells is ten times greater than that of liposomes of comparable size due to their lipid composition and surface proteins, indicating the superior specificity of exosomes for cancer targeting [362]. Furthermore, exosomes offer several advantages over standard delivery vehicles. For example, exosomes are able to cross biological barriers, such as the blood–brain barrier (BBB), have poor immunogenicity and can be cell-specific [363]. Therefore, exosomes could be next-generation nontoxic delivery tools that combine nanoparticle sizes with high capacity levels, making them powerful vectors for the treatment of a variety of pathologies [364].

Doxorubicin-loaded exosomes are transported to tumor tissues and reduce tumor growth in mice without any adverse effects observed from this equipotent free drug [365]. Tian and coworkers used mouse immature dendritic cells (imDCs) for exosome production due to their lack of immunostimulatory markers. Purified imDC-derived exosomes were gently mixed with doxorubicin (DOX) in an electroporation buffer and then examined under a transmission electron microscope to verify the recovery of their plasma membrane. After loading the therapeutic cargo, these vesicles successfully delivered DOX to the targeted cell nucleus, leading to the inhibition of tumor growth without overt toxicity [366]. In another study, exosomes derived from a brain endothelial cell line, bEND.3, were loaded with DOX and used to deliver the anticancer drug across the blood–brain barrier (BBB) for the treatment of brain cancer in a zebrafish model [367]. The membrane vesicles mediated the autonomous intercellular migration of anticancer agents through multiple cancer cell layers and enabled hydrophobic and hydrophilic compounds to significantly penetrate both spheroids and in vivo tumors, thereby enhancing their therapeutic efficacy [368]. Interestingly, chemotherapeutic agents epirubicin and paclitaxel increased miR-503 levels in exosomes released from human umbilical vein endothelial cells (HUVECs) as compared to control conditions and were demonstrated to induce antitumor responses during breast cancer chemotherapy [369].

Exosomes also have the potential to deliver oligonucleotides, such as mRNA, miRNA and various noncoding RNAs, as well as mitochondrial and genomic DNA, to other cells, thus offering considerable advantages as ideal delivery systems for gene therapy [370]. As with the incorporation of genetic material into living cells, Alvarez-Erviti and colleagues used electroporation to deliver short interfering siRNA analogs to the brain in mice via exosomes [363]. In addition, Wahlgren and coworkers used plasma exosomes as gene delivery vectors to transport exogenous siRNA to human blood cells. The vesicles successfully delivered the administered siRNA to monocytes and lymphocytes, leading to robust gene silencing of mitogen-activated protein kinase 1, thus suggesting exosomes as a new generation of drug carriers that enable the development of safe and effective gene therapies [371]. Similarly, Kamerkar et al. demonstrated a technique for the direct and specific targeting of oncogenic KRAS in tumors using electroporated MSC-derived exosomes with siRNA. This treatment suppressed cancer in multiple mouse models of pancreatic cancer and significantly increased overall survival rates [372]. The same method was used to load exosomes with miRNA to the epidermal growth factor receptor (EGFR) expressed in breast cancer cells, indicating that exosomes can be used therapeutically to target EGFR-expressing cancerous tissues with nucleic acid drugs [373]. Finally, endothelial cells treated with chemotherapeutic agents are reported to release more exosomes that contain miRNA-503. Given that miRNA-503 is downregulated in exosomes released from endothelial cells cultured under tumoral conditions, the introduction of miRNA-503 into breast cancer cells altered their proliferative and metastatic capacities by inhibiting both CCND2 and CCND3 [369].

Lee and colleagues demonstrated that exosomes derived from MSCs deliver specific miRNA mimics (miRNA-124 and miRNA-145) and decrease glioma cell migration and the stem cell properties of cancer cells, providing an efficient route of therapeutic miRNA delivery in vivo [374]. In addition, the intratumoral injection of exosomes derived from miRNA-146-expressing MSCs results in a considerable reduction in glioma xenograft development in a rat brain tumor model and decreases cell growth and invasion, suggesting that the export of specific therapeutic miRNA into MSC exosomes represents an effective treatment strategy for malignant glioma [375]. O’Brien and coworkers engineered EVs loaded with miRNA-134, which is substantially downregulated in breast cancer tissue as compared to healthy tissue. It has been demonstrated that miRNA-134-enriched EVs reduce STAT5B and Hsp90 levels in target breast cancer cells, as well as cellular migration and invasion, and enhance the sensitivity of these cancer cells to anti-Hsp90 drugs [376]. Similarly, MSC-derived exosomes encapsulated with miRNA-379 were administered in breast cancer therapy in vivo. The results of this study show that miRNA-379-enriched EVs are potent tumor suppressors with an exciting potential as an innovative therapy for metastatic breast cancer [377]. Bovy et al. identified miRNA-503, whose expression levels are downregulated in exosomes released from endothelial cells cultured under tumoral conditions. Endothelial cells are able to transfer miRNA-503 via exosomes to breast cancer cells, thus impairing their growth and altering their proliferative capacity [369]. Breast cancer cells prime MSCs to secrete exosomes containing distinct miRNA contents, which promotes quiescence in a subset of cancer cells and confers drug resistance. According to this study, a novel therapeutic approach to target dormant breast cancer cells based on the systemic administration of MSCs loaded with antagomiRNA-222/223 resulted in the chemosensitization of cancer cells and increased survival rates [148].

Shtam et al. introduced two different anti-RAD51 and -RAD52 siRNAs into Henrietta Lacks (HeLa) cell-derived exosomes. These exosomes effectively delivered siRNA into the recipient cancer cells and caused strong RAD51 knockdown, providing additional evidence of the ability to use human exosomes as vectors in cancer therapy [378]. In a similar study, Shimbo and coworkers found that the transfer of miRNA-143 by means of MSC-derived exosomes decreases in the in vitro migration of osteosarcoma cells [379]. In addition, miRNA-122-transfected adipose tissue-derived MSCs (AMSCs) can effectively generate miRNA-122-encapsulated exosomes, which can mediate miRNA-122 communication between AMSCs and hepatocellular carcinoma (HCC) cells, thereby elevating tumor cell sensitivity to chemotherapeutic agents through the alteration of miRNA-122 target gene expression in HCC cells [380]. Usman and colleagues have described a strategy for generating large-scale amounts of exosomes for the delivery of RNA drugs, including antisense oligonucleotides (ASOs). They chose human red blood cells (RBCs), which are devoid of DNA, for EV production. RBC EVs were demonstrated to deliver therapeutic ASOs in order to effectively antagonize oncogenic micro-RNAs (oncomiRNAs) and to suppress tumorigenesis [381]. Exosomes could potentially deliver therapeutic proteins to recipient cells, with a recent study reporting the feasibility of using exosomes as biocompatible vectors that could improve the targeting and delivery of therapeutic proteins to specific cells in diseased tissues [382]. In addition, Haney et al. used a new method to treat Parkinson’s disease (PD). In fact, catalase-loaded exosomes produce a potent neuroprotective effect on both in vitro and in mouse brains following intranasal administration. This result demonstrates the capacity of exosomes to load fully functional proteins and to treat specific disorders [383]. Several approaches have envisaged the utilization of specific conserved domains in order to enhance the loading of proteins. For instance, Sterzenbach and colleagues exploited late-domain (L-Domain) proteins and ESCRT machinery pathways to load Cre recombinase into exosomes. This protein was successfully delivered to neurons through a nasal route, a well-characterized noninvasive method to deliver exogenous proteins to the brain via exosomes [384]. Human ubiquitin was also used as a sorting sequence to deliver diverse proteins into exosomes such as EGFP and nHER2. Interestingly, C-terminal–ubiquitin fusion may act as an efficient signal sequence of antigenic proteins into exosomes, which could support the use of exosomes as vaccines [385].

## 5. Conclusions

A considerable number of physiological and pathological processes are undoubtedly governed or, at least, modulated by the intervention of exosomes. This places exosomes in a privileged position and optimizes their use as a potential tool in clinical applications for both diagnosis and therapy. Despite groundbreaking improvements, a number of limitations and challenges remain with regards to transforming exosome applications into clinical therapies. Further exploration of the molecular composition and function of exosomes, along with an appropriate cell source for exosome production according to the intended therapeutic use, will undoubtedly enhance the final outcome of any clinical applications using these membrane vesicles. Taking into account the low biofluid volumes available for diagnosis application, standard and highly effective exosome isolation, purification, characterization and manipulation methods need to be developed to make these vesicles a clinical reality. Furthermore, the loading of exosomes without altering their functional efficacy and the natural characteristics of the donor cell are crucial for oncological research and their development. Finally, with research in exosome biology in its infancy, further studies to evaluate the possible impacts of exosomes in major preclinical models are required to assess the safety/toxicology issues and to ensure their safe and effective use in therapeutic settings.

## Figures and Tables

**Figure 1 cancers-13-00084-f001:**
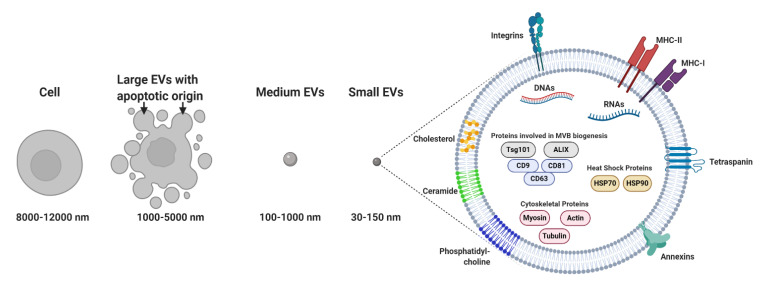
Sizes of most common cell particles: small extracellular vesicles (EVs) are 30 nm to 150 nm in size, and medium-sized EVs are in the 100 nm to 1000 nm range, while large EVs of apoptotic origin are typically 50 nm to 5000 nm in diameter [11]. The release of small EVs or exosomes differs from migracytosis, which involves the translocation of cytoplasmic material to migrasomes, followed by their release when the retraction fibers break [12]. In addition, the uptake of small EVs may have an effect on recipient cells different from that of multivesicular body (MVB)-like EVs, whose release may lead to a relatively delayed effect on the microenvironment [13]. Molecular composition of exosomes: exosomes are surrounded by a phospholipid bilayer and contain numerous molecules, including proteins, lipids, DNA and several types of RNA. (MHC: major histocompatibility complex).

**Figure 2 cancers-13-00084-f002:**
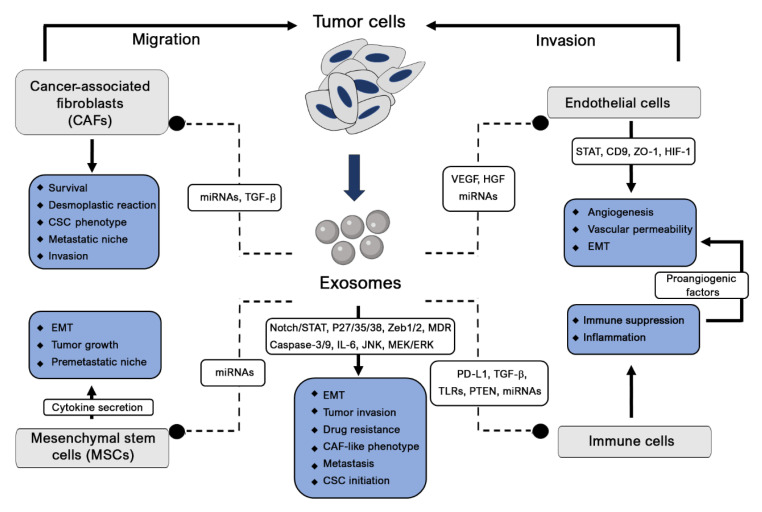
Roles of tumor-derived exosomes in cancer pathogenesis. Cancer stem cells: CSC. Epithelial-to-mesenchymal transition: EMT. microRNAs: miRNAs. Transforming growth factor beta: TGF-β. Signal transducer and activator of transcription: Stat. zinc finger E-box-binding homeobox: Zeb. Multidrug resistance: MDR. Interleukin 6: IL-6. Jun N-terminal kinases: JNK. Mitogen-activated protein kinase kinase: MEK. Extracellular signal-regulated kinases: ERK. Vascular endothelial growth factor: VEGF. Hepatocyte growth factor: HGF. Programmed death-ligand 1: PD-L1. Toll-like receptors: TLRs. Phosphatase and tensin homolog: PTEN. Cluster of differentiation 9: CD9. Zonula occludens-1: ZO-1. Hypoxia inducible factor-1: HIF-1).

**Table 2 cancers-13-00084-t002:** Representative table showing a selection of concluded and ongoing clinical trials (Clinical trial.gov; November 2020) utilizing exosomes as biomarkers mainly for cancer diagnosis and, to a lesser extent, for the early detection of other diseases. The clinical work involving the diagnostic application of exosomes has not yet been published or made available for peer review.

CT Identification	Aim of Study	Source of Exosomes	Associated Markers	Promoted by
NCT04182893	Identification of benign and malignant pulmonary nodules	Blood and alveolar lavage fluid	Exosomal RNA	Shanghai Chest Hospital, Shanghai, China
NCT04499794	Study of exosome EML4-ALK fusion in NSCLC clinical diagnosis	Plasma	EML4-ALK fusion	Cancer Hospital Chinese Academy of Medical Sciences, Beijing, China
NCT03032913	Onco-exosome quantification in diagnosis of pancreatic cancer	Blood	Onco-exosomes	CHU de Bordeaux, Bordeaux, France
NCT04529915	Early diagnosis of lung cancer using blood plasma-derived exosomes	Blood	Exosomal proteins	Korea University Guro Hospital, Seoul, Republic of Korea
NCT04394572	Identification of new diagnostic protein markers for colorectal cancer	Blood	Exosomal proteins	CHU Reims, Reims, France
NCT04155359	Diagnosis of bladder cancer in hematuria patients	Urine	sncRNAs	Integrated Medical Professionals, Farmingdale, New York, United States
NCT03974204	Analysis of exosomes in cerebrospinal fluid for breast cancer patients	Cerebrospinal fluid and blood	Exosomal proteins	Centre Hospitalier Régional Universitaire de Lille, Lille, Hauts-de-France, France
NCT03830619	Exosomal long noncoding RNAs as potential biomarkers for lung cancer diagnosis	Plasma	Exosomal lncRNAs	Union Hospital, Tongji Medical College, Huazhong University of Science and Technology, Wuhan, Hubei, China
NCT03562715	Role of exosomal miRNAs 136, 494 and 495 in pre-eclampsia diagnosis	Blood	miRNAs 136, 494 and 495	Cairo University, Cairo, Egypt
NCT03415984	Estimation of age-related macular degeneration (ARMD) prevalence in Parkinson’s patients	Not defined	Pro-inflammatory components	Fondation Ophtalmologique Adolphe de Rothschild, Paris, France
NCT04523389	Analysis of extracellular vesicle contents as biomarkers in colorectal cancer patients	Blood	miRNAs	CHU Dijon Bourgogne, Dijon, France
NCT03108677	Evaluation of circulating exosomal RNA as biomarker for lung metastases of primary high-grade osteosarcoma	Blood	Exosomal RNA	Ruijin Hospital, Shanghai, China
NCT04258735	Genomic analysis of metastatic breast cancer patients	Blood	ctDNA and RNA	Sungkyunkwan University School of Medicine, Seoul, Republic of Korea
NCT04053855	Evaluation of urinary exosomes presence from clear cell renal cell carcinoma	Urine	CD9,CD63,CD81,CA9 and VGEFR2	CHU Saint-Etienne, Saint-Étienne, France
NCT04315753	assessment of exosomes’ role in improving lung cancer management and early detection	Blood	Exosomal antigens	Istituto Clinico Humanitas, Rozzano, Milano, Italy
NCT04459182	evaluation of miRNA in exosomes in obese and OSA patients with endothelial dysfunction	Not defined	miRNA	University Hospital, Angers, France
NCT04556916	Early detection of prostate cancer	Blood	Exosomes	University Hospital, Montpellier, France
NCT02464930	Evaluation of microRNA expression in blood and cytology for detecting Barrett’s esophagus and associated neoplasia	Bile and serum	miRNAs 192-5p, 215-5p and 194-5p	Kansas City Veterans Affairs Medical Center, Kansas City, Missouri, United States
NCT03800121	Study of exosomes in monitoring patients with sarcoma	Blood	Exosomal RNA and proteins	Centre Georges François Leclerc, Dijon, France
NCT04154332	Defining the functional role of exosomes in the development of preeclampsia leading to cardiovascular remodeling	Urine and Blood	Exosome abnormalities	University of Alabama,, Birmingham, Alabama, United States
NCT03102268	Characterization of cholangiocarcinoma-derived exosomal ncRNAs	Plasma	Non-coding RNAs	Second Affiliated Hospital of Nanjing Medical University, Nanjing, Jiangsu, China
NCT03581435	Study of circulating exosome proteomics in gallbladder carcinoma patients	Blood	Exosomal proteins	Xinhua Hospital, Shanghai, China
NCT03738319	Analysis of non-coding RNAs in epithelia ovarian cancer	Blood	miRNA and lncRNA	Peking Union Medical College Hospital, Beijing, China
NCT03911999	Investigating relationship between urinary exosomes and aggressiveness of prostate cancer	Urine	Exosomal miRNA	Prince of Wales Hospital, Hong Kong, Hong Kong
NCT04120272	Search for biomarkers for early detection and prevention of delirium	Urine and blood	miRNA	College of Nursing, Yonsei University, Seoul, Republic of Korea
NCT03419000	Evaluation of microRNAs as biomarkers of respiratory dysfunction in refractory epilepsy	Blood	miRNAs	Hospices Civils de Lyon, Bron, France
NCT04534647	Assessment of correlation between serological and urinary biomarkers and systemic lupus erythematosus	Blood and urine	Exosomes	Liga Panamericana de Asociaciones de Reumatologia (PANLAR), Rosario, Argentina
NCT02147418	Exosome testing as screening modality for human papillomavirus-positive oropharyngeal squamous cell carcinoma	Saliva	Exosomal proteins	New Mexico Cancer Care Alliance, Albuquerque, New Mexico, United States

## Data Availability

The data presented in this study are available on request from the corresponding authors.

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
