# Peer review of "Exosomes: Their Role in Pathogenesis, Diagnosis and Treatment of Diseases"

_cancers, 2020, doi:10.3390/cancers13010084_

Round 1
Reviewer 1 Report
Aheget and colleagues intended to review several aspects of exosomes in the pathogenesis and treatment of disease, which looks like a very or, rather, too ambitious challenge. Nevertheless, their effors are appreciated and encouraged after considering the critical points below.
The title should be more specific, and at least the diagnostic aspect, which is probably the most important to date, should be included.
It is not very clear why they picked up neurodegenerative and renal disease out of the spectrum shown in Table 2. It must be at least explained.
The reveiw is mainly decriptive by piling up a long list of published data usually without the critical views by the authors, which would have made this paper more vivid.
Description of the molecular make up of the small extracellular vesicles is too long on the expense of sparing on the rest.
For subtyping extracellular vesicles the ISEV concensus nomenclature should be used i.e. small-, medium-, large EVs. (DOI: 10.1080/20013078.2018.1535750).
Accordingly „apoptotic bodies” should be replaced to „large EVs with apoptotic origin”.
The relation between exosomes and small EV should also be clarified since small EV may be released from other sources e.g. from migrasomes (DOI: 10.1038/cr.2014.135) or MVB-like EV clusters (DOI: 10.1080/20013078.2019.1596668). Though we are aware of the published deviations in exosome sizes, but 150 nm for an upper limit would be wiser (DOI: 10.1172/JCI81135).
Line 60. indication of genomial DNS is missing (DOI: 10.1038/cr.2014.44), particularly because it is mentioned in line 733.
Line 76. It would be useful to start listing of EV content in the order of their frequency of occurence in EVs.
Line 186. DNA is missing from the chapter discussing nucleic acids, why? It should be at least explained.
Line 231. The cancer promoting effect of exosomesis not limited only to miRNA.
Line 260. The major points/stages of cancer initiation should be briefly described.
Line 333. Tumor angiogenesis appear here again despite it is already discussed in the previous chapter. They should be merged.
Lines 362-380. In cancer drug resistance they focus only on miRNAs, other basic mechamisms are missing e.g. tumor-stroma interactions (DOI: 10.1186/s12943-019-0975-5, 10.1007/s10555-013-9441-9, 10.3390/cancers12082324.)
Line 396. Confusing statement: …and the design of advances strategies in the treatment….
Line 518. Exosomes in cancer diagnostics… DNA is missing, why? Explain or include!
Line 568. The abbreviation of miRNA should be consolidated, e.g. it is miR in line 487.
Line 658. Kidney diseases suddenly occur, despite it was not mentioned before, why? It should be explained.
Lines 704-707 and 732-734. A brief statement on how the contents of exosomes are filled in.
Author Response
Reviewer 1
- The title should be more specific, and at least the diagnostic aspect, which is probably the most important to date, should be included.
Thank you for your comment, we have modified the title and included the diagnostic aspect.
- It is not very clear why they picked up neurodegenerative and renal disease out of the spectrum shown in Table 2. It must be at least explained.
Thank you for your comment; we have added the only study that we found regarding the diagnostic application of exosomes in neurodegenerative diseases. Renal disease was removed from the review.
- Description of the molecular make up of the small extracellular vesicles is too long on the expense of sparing on the rest.
Thank you for your comment; we have eliminated some references in each chapter.
- For subtyping extracellular vesicles the ISEV consensus nomenclature should be used i.e. small-, medium-, large EVs. (DOI: 10.1080/20013078.2018.1535750).
Thank you for your comment; we have used the ISEV consensus nomenclature for subtyping extracellular vesicles (Figure 1, Line: 49).
- Accordingly„ apoptotic bodies” should be replaced to „large EVs with apoptotic origin”.
Thank you for your comment, we have used “large EVs with apoptotic origin” instead of “apoptotic bodies” (Figure 1, Line: 50).
- The relation between exosomes and small EV should also be clarified since small EV may be released from other sources e.g. from migrasomes (DOI: 10.1038/cr.2014.135) or MVB-like EV clusters (DOI: 10.1080/20013078.2019.1596668).Though we are aware of the published deviations in exosome sizes, but 150 nm for an upper limit would be wiser (DOI: 10.1172/JCI81135).
Thank you for your comment, we have clarified the relation between exosomes and small EV and compared them to migrasomes and MVB-like EV clusters (Line: 50-54).
We have used 150 nm for upper limit (Figure 1, Line: 49).
- Line 60. indication of genomic DNA is missing (DOI: 10.1038/cr.2014.44), particularly because it is mentioned in line 733.
Thank you for your comment, we have included genomic DNA (Line: 458).
- Line 76. It would be useful to start listing of EV content in the order of their frequency of occurence in EVs.
Thank you for your comment; we have listed EV content in the order of their frequency of occurence in EVs and in accordance with the diagnosis chapter (Line: 470).
- Line 186. DNA is missing from the chapter discussing nucleic acids, why? It should be at least explained.
Thank you for your comment; we have added DNA in the chapter discussing nucleic acids (Line: 472 -475).
- Line 231. The cancer promoting effect of exosomes is not limited only to miRNA.
Thank you for your comment, we have discussed other cancer initiation factors (Line:79).
- Line 260. The major points/stages of cancer initiation should be briefly described.
Thank you for your comment, we have described the major points/stages of cancer initiation (Line: 124).
- Line 333. Tumor angiogenesis appear here again despite it is already discussed in the previous chapter. They should be merged.
Thank you for your comment; we have merged the tumor angiogenesis sentences (Line: 177).
- Lines 362-380. In cancer drug resistance they focus only on miRNAs, other basic mechamisms are missing e.g. tumor-stroma interactions (DOI: 10.1186/s12943-019-0975-5, 10.1007/s10555-013-9441-9, 10.3390/cancers12082324.)
Thank you for your comment, we have added the tumor-stroma interactions and other mechanisms in the cancer drug resistance chapter (Line: 292).
- Line 396. Confusing statement: …and the design of advances strategies in the treatment…
Thank you for your comment, we have corrected it to “could play a role in advanced strategies of treatments of neurodegenerative diseases” (Line: 369).
- Line 518. Exosomes in cancer diagnostics… DNA is missing, why? Explain or include!
Thank you for your comment, we have discussed DNA as well in cancer diagnostics (Line: 608).
- Line 568. The abbreviation of miRNA should be consolidated, e.g. it is miR in line 487.
Thank you for your comment, we have consolidated the abbreviation of miRNA in all the manuscript.
- Line 658. Kidney diseases suddenly occur, despite it was not mentioned before, why? It should be explained.
Thank you for your comment, we have deleted the kidney diseases paragraph, and detailed more the neurodegenerative one (Line: 732).
- Lines 704-707 and 732-734. A brief statement on how the contents of exosomes are filled in.
Thank you for your comment, we have tried to explain how the content of exosomes is filled in, even though it is not well explained in original papers (Line: 774-776 and 797-798).

Reviewer 2 Report
Current Knowledge on Exosomes: Their Role in Pathogenesis and Treatment of Diseases , by Houssam Aheget et al
First of all I should say that being the title of this journal “Cancers”, this review is more suitable for a generalist journal in science. For instance of the same publisher International Journal of Molecular Science.
Then a very recent article has been published In cancers dealing with exosomes and cancer in a translational way whose title is “Exosomes: A Source for New and Old Biomarkers in Cancer” by Logozzi M. et al (Cancers 2020, doi: 10.3390/cancers12092566)
Of course the issue of the role of exosomes in cancer biology, diagnosis and treatment is very timely. However, this reviewer does not believe there are so many data available in order to distinguish a cancer from another through exosomes cargo. All the attempts to present new specific tumor biomarkers as shuttled by exosomes is plus or minus failed, and this is unfortunately true not only for scientific research but from industrial R & D as well.
However, this reviewer believes that cancers have many common phenotypes, including microenvironmental acidity and hypoxia and low nutrient supply, and more recently an increased exosome release as well. The latter has been shown in some clinical studies as well, that included melanoma, brain tumors, oral cancers and prostate cancer patients. Moreover, a recent clinical study has shown that also well known tumor biomarkers, such PSA, when detected on plasmatic exosomes can distinguish prostate cancer patients from both BPH and healthy individuals. Thus, a possiblility to use the number of plasmatic exosomes for a clinical follow up of tumor patients looks mandatory.
Exosomes are for sure probably the most important discovery of the last decades and for sure they will provide further and very important new and milestones discoveries in both the scientific knowledge and medicine, but we should manage to realize that research in medicine is hugely different from the simply knowledge increasing. As an example recent report have shown that exosome may deliver genomic information to the germ line in vivo, thus changing the Watson and Crick pardigm and probably providing the real way our genome was filled up of retroelement during its filoontogenesis.
Lastly the authors should discuss missing issues including the ability of exosomes to increase drug resistance; the ability of exosomes to deliver antitumor drugs; the ability of exosomes to deliver proteins with a full functionality, such as carbonic anhydrase; the evidence that plasmatic exosomes from cancer patients are acidic as well.
Author Response
Authors should discuss missing issues including:
- The ability of exosomes to increase drug resistance
Thank you for your comment, we have discussed the ability of exosomes to increase drug resistance (Line: 292).
- The ability of exosomes to deliver antitumor drugs
Thank you for your comment, we have discussed the ability of exosomes to deliver antitumor drugs (Line: 772).
- The ability of exosomes to deliver proteins with a full functionality, such as carbonic anhydrase;
Thank you for your comment, we have not found any article that discuss the delivery of carbonic anhydrase via exosomes, please let us know if there are some. Meanwhile, we have discussed the ability of exosomes to deliver therapeutic proteins and proteins with full functionality just before the conclusion (Line: 830).
- The evidence that plasmatic exosomes from cancer patients are acidic as well.
Thank you for your comment, we have discussed the evidence that plasmatic exosomes from cancer patients are acidic (Line: 701).

Reviewer 3 Report
Current Knowledge on Exosomes: Their Role in Pathogenesis and Treatment of Disease
In this well written manuscript by Aheget and colleagues they described exosome biology and their roles in cancer and neurodegenerative diseases. The utility of exosomes as a diagnostic and delivery tool for the treatment of these pathological conditions was discussed. Specific comments are:
- Corrections in English are required to improve the readability of this manuscript.
- Line 29 remove “delimited”.
- Line 45 instead of “formed by a series of…” do you mean “composed of molecules…”?
- Line 44-48 should be broken into two sentences. One sentence should be about microvesicles and the other one should be about exosomes since it’s a run-on sentence.
- Figure 1 make the units the same. Make um into nm in order to easily compare sizes. In addition to Tsg101 and Alix add CD63, CD9 and CD81.
- Exosomes size range is 30 nm – 120 nm. Please change 100 nm to 120 nm.
- Line 217 define yRNA.
- Line 270, the quotes, “suppressor of fused” is confusing. Please delete or explain in the text.
- In section 3.1.3 please describe in detail pre-metastatic niche formation by exosomes and the “seed and soil hypothesis” in respect to exosomes.
- A figure describing section 3 should be included.
- In section 3.1.4 talk about exosomal PD-L1 expression.
Chen, G.; Huang, A.C.; Zhang, W.; Zhang, G.; Wu, M.; Xu, W.; Yu, Z.; Yang, J.; Wang, B.; Sun, H.; et al. Exosomal PD-L1 contributes to immunosuppression and is associated with anti-PD-1 response. Nature 2018, 560, 382–386. [Google Scholar] [CrossRef] [PubMed]
Poggio, M.; Hu, T.; Pai, C.C.; Chu, B.; Belair, C.D.; Chang, A.; Montabana, E.; Lang, U.E.; Fu, Q.; Fong, L.; et al. Suppression of Exosomal PD-L1 Induces Systemic Anti-tumor Immunity and Memory. Cell 2019, 177, 414–427.e413. [Google Scholar] [CrossRef] [PubMed]
- Line 558, why is cancer italicized?
- Line 567 what is ECSS, it is not defined?
- Why is the first sentence in figure 2 underlined?
Author Response
- Corrections in English are required to improve the readability of this manuscript.
Thank you for your comment, we have sent the review to proofreading services. They are going to revise it again to cover all the new modifications.
- Line 29 remove “delimited”.
Thank you for your comment, we have removed “delimited” (Line: 27).
- Line 45 instead of “formed by a series of…” do you mean “composed of molecules…”?
Thank you for your comment, we have replaced “formed by a series of” by “composed of molecules” (Line: 40).
- Line 44-48 should be broken into two sentences. One sentence should be about microvesicles and the other one should be about exosomes since it’s a run-on sentence.
Thank you for your comment, the proofreading service has suggested this new modification (Line: 39). Please let us know if it needs more correction.
- Figure 1 make the units the same. Make um into nm in order to easily compare sizes. In addition to Tsg101 and Alix add CD63, CD9 and CD81.
Thank you for your comment, we have made the units the same “um into nm” (Figure 1, Line: 49).
We have added CD63, CD9 and CD81 to Tsg101 and Alix in the figure.
- Exosomes size range is 30 nm – 120 nm. Please change 100 nm to 120 nm.
Thank you for your comment, another reviewer has suggested 150 nm as upper limit.
We changed it to 150 nm since it includes both (Line: 49).
- Line 217 define yRNA.
Thank you for your comment, we did not find a definition for yRNA, please let us know if there is one.
- Line 270, the quotes, “suppressor of fused” is confusing. Please delete or explain in the text.
Thank you for your comment, we have eliminated “suppressor of fused” (Line: 147).
- In section 3.1.3 please describe in detail pre-metastatic niche formation by exosomes and the “seed and soil hypothesis” in respect to exosomes.
Thank you for your comment, we have described in detail pre-metastatic niche formation by exosomes and the “seed and soil hypothesis” in respect to exosomes (Line: 184).
- A figure describing section 3 should be included.
Thank you for your comment, we have included a figure describing that section (Line: 103).
- In section 3.1.4 talk about exosomal PD-L1 expression.
Chen, G.; Huang, A.C.; Zhang, W.; Zhang, G.; Wu, M.; Xu, W.; Yu, Z.; Yang, J.; Wang, B.; Sun, H.; et al. Exosomal PD-L1 contributes to immunosuppression and is associated with anti-PD-1 response. Nature 2018, 560, 382–386. [Google Scholar] [CrossRef] [PubMed]
Poggio, M.; Hu, T.; Pai, C.C.; Chu, B.; Belair, C.D.; Chang, A.; Montabana, E.; Lang, U.E.; Fu, Q.; Fong, L.; et al. Suppression of Exosomal PD-L1 Induces Systemic Anti-tumor Immunity and Memory. Cell 2019, 177, 414–427.e413. [Google Scholar] [CrossRef] [PubMed]
Thank you for your comment, we have talked about exosomal PD-L1 expression in that section (Line: 263).
- Line 558, why is cancer italicized?
Thank you for your comment, we have removed italic type from the word “cancer” (Line: 637).
- Line 567 what is ECSS, it is not defined?
Thank you for your comment, we have corrected the word ESCC (Line: 647).
- Why is the first sentence in figure 2 underlined?
Thank you for your comment, we have removed the line under the first sentence in figure 2 (Line: 746).

Round 2
Reviewer 1 Report
The authors carfully corrected all suggested critical points in their review. The only suggestion left for the reviewer is to clarify the sentence in
Lines 124-125, e.g. as "Cancer results from the irreversible genetic deregulation in cells due to activation of specific oncogenes, inactivation of tumor-suppressive genes or other genes involved in genome stability."
Reviewer 2 Report
IN GENERAL THE MANUSCRIPT IS IMPROVED BUT I INSIST THAT THIS PAPER IS NOT SUITABLE FOR PUBLICATION IN CANCERS BECAUSE IT DOES NOT DEAL SPECIFICALLY WITH CANCER